behaviour/computational biology/biophysics

*Leptothorax*, ants, multi-rhythmicity, excitable media, coupled oscillators

**Author for correspondence:**
Grant Navid Doering
e-mail: naviddio@gmail.com

# Noise resistant synchronization and collective rhythm switching in a model of animal group locomotion

Grant Navid Doering[1], Brian Drawert[3], Carmen Lee[2], Jonathan N. Pruitt[1], Linda R. Petzold[4,5] and Kari Dalnoki-Veress[2]

[1]Department of Psychology, Neuroscience and Behaviour, and [2]Department of Physics and Astronomy, McMaster University, Hamilton, Ontario, Canada L8S 4K1
[3]National Environmental Modeling and Analysis Center, University of North Carolina at Asheville, Asheville, NC 28804, USA
[4]Department of Computer Science, and [5]Department of Mechanical Engineering, University of California, Santa Barbara, CA 93106, USA

GND, 0000-0002-3839-9139; JNP, 0000-0003-0326-6755

Biology is suffused with rhythmic behaviour, and interacting biological oscillators often synchronize their rhythms with one another. Colonies of some ant species are able to synchronize their activity to fall into coherent bursts, but models of this phenomenon have neglected the potential effects of intrinsic noise and interspecific differences in individual-level behaviour. We investigated the individual and collective activity patterns of two *Leptothorax* ant species. We show that in one species (*Leptothorax* sp. W), ants converge onto rhythmic cycles of synchronized collective activity with a period of about 20 min. A second species (*Leptothorax crassipilis*) exhibits more complex collective dynamics, where dominant collective cycle periods range from 16 min to 2.8 h. Recordings that last 35 h reveal that, in both species, the same colony can exhibit multiple oscillation frequencies. We observe that workers of both species can be stimulated by nest-mates to become active after a refractory resting period, but the durations of refractory periods differ between the species and can be highly variable. We model the emergence of synchronized rhythms using an agent-based model informed by our empirical data. This simple model successfully generates synchronized group oscillations despite the addition of noise to ants' refractory periods.

We also find that adding noise reduces the likelihood that the model will spontaneously switch between distinct collective cycle frequencies.

## 1. Introduction

Synchronization is one of the most pervasive examples of collective behaviour, being present throughout numerous biological [1] and physical contexts [2]. An extensive literature exists on the synchronization of coupled oscillators [3], and many fundamental aspects of synchronization are consequently well understood. More recently, however, efforts have shifted towards understanding the generation of rhythms and synchronization in more complex situations. Chief among these are scenarios that involve mobile oscillators [4], heterogeneity [5] and the role of noise in synchronization [6]. These features are especially relevant to the study of synchronized behaviour in animal social groups because they frequently mingle all three elements, having constituents that are mobile, inherently noisy and heterogenous in their behaviour [7–9].

Insect societies provide an excellent opportunity to experimentally investigate social synchronization because, in some taxa, the entire population of a colony can be observed simultaneously, and the behaviours of separate individuals can be directly assessed [10]. Several species of ants exhibit reliable short-term activity cycles (STACs), where worker ants inside a nest partition their activity into coherent, repeating pulses with periods ranging from 20 to 50 min [11–13]. Colony tasks, like trophallaxis or feeding larvae, are believed to be fulfilled during these activity bursts, as most ants remain motionless during the time separating cycles [14]. Ant STACs are generated endogenously; there is no evidence for any kind of external signal that synchronizes colonies [12], and although the presence of a queen can help to maintain STACs, neither she nor any other specific ant is necessary for these activity cycles to emerge [12,13]. Individual worker ants can move and become active through their own agency in an arrhythmic fashion [15] but can also stimulate nest-mates to become active [16]. Activity pulses can therefore propagate through the colony analogous to a wave [17].

Most previous studies on ant STACs have been conducted using colonies from the closely related genera *Temnothorax* and *Leptothorax* [12]. These genera often have simple, single-chambered nests [18] with small colonies (less than 200 workers) where all individuals can be monitored continuously. Although some work has been directed at modelling periodic activity waves in ants [19,20], empirical data are scarce. Moreover, several aspects of the physics underlying this phenomenon are not understood [21]. For instance, it is not known how noise in the behaviour of individual ants may alter their synchronization. In this context, we define noise as the amount of inherent randomness or unpredictability in the behaviour of individuals. Noise, defined in this way as probabilistic behaviour, is pervasive in biology [22] and can be essential to the spatio-temporal characteristics of coupled oscillators and excitable media [23]. In the phenomenon of coherence resonance, for example, a group of oscillators that share a single external source of noise can experience greater levels of synchronization than they would without noise [6].

There is evidence that worker ants are likely to have refractory periods where they are inactive and less susceptible to activation by nest-mates [16,24]. Because many individual-level behaviours in ants are probabilistic and are not rigidly predictable [25,26], the durations of these refractory periods are not expected to be absolute [24]. The lengths of time that workers are refractive are instead likely to fluctuate randomly for each ant within some range. Different species also appear to oscillate in distinct frequency ranges [13,14,17], and it has been argued that colonies appear to be capable of exhibiting multi-rhythmicity [12], which is defined as a spontaneous switching between different oscillation frequencies [27]. Models of ant STACs have yet to tackle the possible causes of interspecific differences in cycle frequency, the potential for STAC multi-rhythmicity or the effects of noise in ants' refractory periods. It is plausible that there are interspecific differences in individual-level behaviour that account for the variation seen in STAC frequencies between species. Like other models of excitable media [6,28], it is also conceivable that when intrinsic behavioural noise is added to STAC models, the rhythms of collective oscillation may become more predictable. We sought to address these topics by first conducting a set of exploratory observations with colonies and individuals from two previously unstudied Nearctic species of *Leptothorax*: *Leptothorax crassipilis* (figure 1*a*) and the taxonomically undescribed *Leptothorax* sp. W (figure 1*c*). Using these empirical observations, we then built an agent-based model of STACs and investigated whether (i) collective-level interspecific differences in STAC frequencies could be reclaimed by our model, (ii) if collective oscillations can

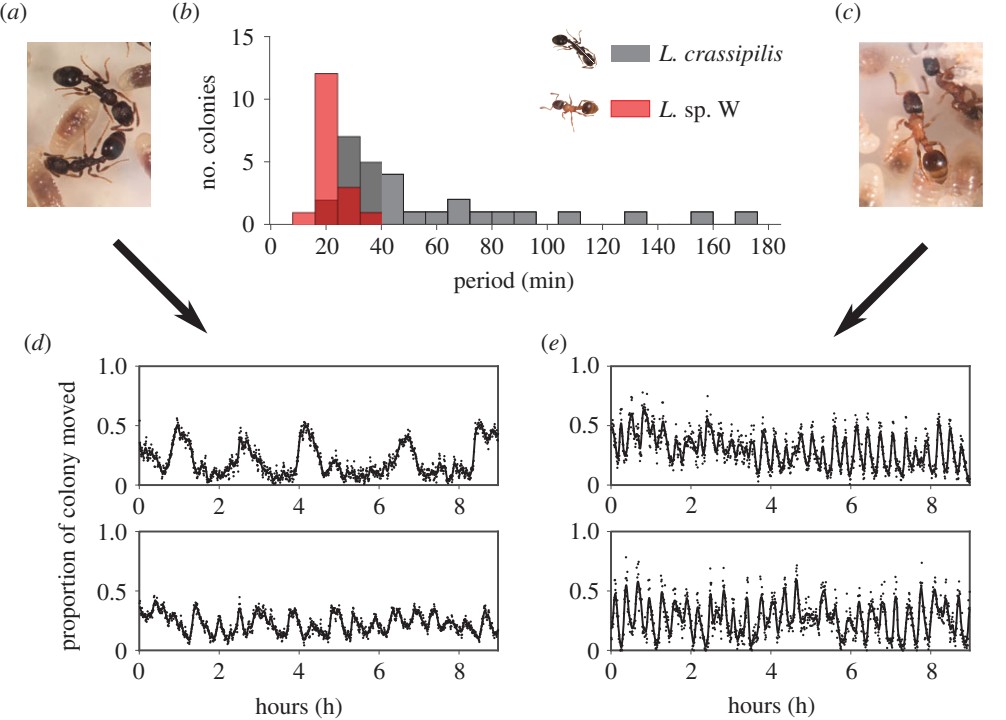

**Figure 1.** Empirical data from colonies. (*a*) *Leptothorax crassipilis* workers with brood. (*b*) Histograms of STAC periods expressed by colonies of both *Leptothorax* species. (*c*) *Leptothorax* sp. W workers with brood. (*d*) Time series of collective locomotor activity from two representative *L. crassipilis* colonies and (*e*) two *Leptothorax* sp. W colonies. Black dots in time series represent unprocessed data points, and lines show the smoothed weighted moving average. *Leptothorax* sp. W colonies show little variation in collective frequencies, but *L. crassipilis* can express a range of cycle frequencies.

survive in the presence of stochastic refractory periods, and (iii) if refractory noise can induce or inhibit multi-rhythmic oscillations.

## 2. Material and methods

### 2.1. Colony information

The *L. crassipilis* Wheeler 1917 colonies used in this study were collected from rock crevices in the Pinal Mountains near Globe, Arizona, in February and May 2018 and June 2019. The *Leptothorax* sp. W colonies were collected from rotting acorns in Fish Creek, Wisconsin, in July 2018 and in May and July 2019. *Leptothorax crassipilis* colonies ranged in size from 8 to 248 individuals (mean: 73.5, s.d.: 52.4), and *Leptothorax* sp. W colony sizes spanned 7–61 individuals (mean: 23.3, s.d.: 17.5). Six brood-less *Leptothorax* sp. W colony fragments with less than five workers each contributed ants for our studies on isolated workers. Colonies were maintained using standard ant husbandry techniques (electronic supplementary material).

### 2.2. Activity measurements

We filmed 23 colonies of *L. crassipilis* and 15 colonies of *Leptothorax* sp. W for approximately 9 h each to characterize the typical patterns of collective movement activity in both species. Additionally, we haphazardly selected two colonies of *Leptothorax* sp. W and four colonies of *L. crassipilis* (plus two additional *L. crassipilis* colonies not from the original set of 23) to be filmed for 35 h to examine how cycles change over a longer observation window. The colonies used for these longer recordings were all collected in 2018 as we filmed the long recordings prior to our acquisition of the colonies that were collected in 2019.

Time series of collective locomotor activity for entire colonies were obtained using a version [13] of the automated techniques originally developed by Cole [14], Hatcher [24] and Tofts *et al.* [29].

Colonies' nest-boxes (11 × 11 × 3 cm) were placed over pink/white paper to enhance contrast with the ants and recorded with Canon VIXA camcorders. Colony recordings were processed by extracting frames from each video to generate image sequences where each image was separated from the next by 30 s. Each image in a sequence was binarized using an adaptive threshold [30], so that all objects other than ants residing in their nest were filtered out of the image. Regions in an image that contain ants can be distinguished from non-ants due to the insects' dark integument appearing over the lighter paper background. Pairs of successive images were then subtracted from each other to determine the number of pixels that had changed from 0 to 1 between frames, and this quantity was divided by the number of pixels in the first frame of each pair to estimate the proportion of ants in a nest that moved every 30 s [13,17].

We studied the movements of isolated ants to see if individual-level behavioural patterns differed between the two species and to guide the parametrization our agent-based model. Previous work in *Temnothorax allardycei* using isolated workers and small groups of ants removed from their nests has shown that STACs emerge gradually as aggregate size is increased [31]. This result suggests that studying workers in isolation can provide at least some insight into the mechanisms that enable STACs in colonies. Twenty workers from each species were removed from multiple source colonies, and each individual was filmed in isolation for 30.8 h, so that movement patterns could be tracked in the absence of social interactions. Recordings of isolated individual ants were conducted by confining workers to separate plastic Petri dishes (45 mm diameter). The cotton tip of a tube of water was available to ants in each dish through a hole drilled in the side of each dish. A damp cotton plug blocked escape through the hole and provided the ants with a constant source of moisture to prevent desiccation over long filming sessions. One *L. crassipilis* worker was injured and perished while it was being isolated, resulting in one fewer individual-level time series for that species. Because recordings of isolated singletons involved only one ant in each video, we automatically tracked the locomotor activity (confined to two dimensions) of these individuals by calculating the distance the centroid of the focal ant moved in pixels every 30 s [13].

## 2.3. Time-series analysis

All empirical time series examined in this study were analysed in the same way. Time series were first processed with a Gaussian-weighted moving average filter with a window size of 15 points (i.e. 7.5 min) to reduce noise. Smoothing the time series with this window size prevented the spurious detection of extremely fast oscillations that were merely artefacts of the tracking algorithm (electronic supplementary material, figure S1). Data were then normalized, so that the largest and smallest values in a time series were reassigned to be 1 and 0 respectively, and all intermediate values were rescaled to fall between these two points. The locations of peaks in activity time series were determined using the Matlab function *findpeaks*. This function was set to detect peaks in the time series that exceeded a prominence of 0.2 units of normalized activity [13]. These automatically detected peak locations were used to compute the mean inter-beat interval (IBI) and coefficient of variation (CV) associated with each time series. The CV was defined as the variability (standard deviation) in time between automatically detected activity peaks ($T_p$) divided by the mean time between peaks (i.e. the mean IBI).

$$\mathrm{CV} = \frac{\mathrm{std}(T_p)}{\langle T_p \rangle}, \tag{2.1}$$

For time series of colony-level activity, we calculated the dominant oscillation period of each smoothed and rescaled time series using wavelet analysis, which is well suited to process the often non-stationary activity patterns of ant colonies [13,32]. The wavelet analyses to detect the dominant periods in colony activity time series were conducted in Matlab using a one-dimensional Morse continuous wavelet transform implemented with the *cwt* function. Briefly, after computing the continuous wavelet transform of each colony time series, we excluded results occurring within the 'cone-of-influence' to reduce edge artefacts. We then found the frequency band associated with the highest wavelet magnitude. It should be noted that this method can result in identical estimates of period for different time series. Previous work provides greater detail about using this method on ant activity cycles [13]. Like their empirical counterparts, time series obtained from all model simulation runs were also processed with a 15-point moving average filter before we applied wavelet analysis. Because the long simulation outputs from our agent-based model exhibited stationarity, we also used Lomb–Scargle spectral analysis on these time series. All time-series summary data are presented as average ± s.d.

In addition to the wavelet analysis described above, the 35 h recordings of colony activity were also analysed with Lomb–Scargle periodograms to explore whether colonies could exhibit different oscillation frequencies within the same time series. Because the 35 h time series are somewhat non-stationary, we detrended the 35 h time series prior to Lomb–Scargle spectral analysis to remove trends in the data that were not part of STACs. This was done using the *detrend* function in Matlab with a fourth-degree polynomial. Because these time series are 35 h long, sustained oscillations from STACs should be detectable in the power spectra.

## 2.4. Ant–ant interactions

Physical encounters between individual ants can promote activity in dormant individuals and spread activity throughout *Leptothorax* nests [16,24,33]. Because physical touch spreads activity in these ants, interspecific differences in how ants respond to encounters may also exist between the two *Leptothorax* species, which in turn could affect their collective activity cycles. Acquiring empirical information on how workers in both species react to physical stimulation is also necessary to inform the construction of our agent-based model. We therefore collected data on the activity patterns of ants when they were among their sisters inside their nests. First, we investigated the likelihood that inactive ants would respond to physical interactions with their nest-mates. We randomly selected (haphazardly, without the aid of a pseudorandom number generator) video recordings of two colonies of each species and selected 15 focal ants from each video that became active through stimulation during a single, predetermined cycle of colony activity. We recorded the times at which any ant made tactile contact with the inactive focal ants, and whether contact elicited activity from the focal ants (see electronic supplementary material). To confirm that refractory-like periods are indeed present in both species, we used binomial generalized linear mixed models (GLMMs) to test whether there was a relationship between an ant's length of time inactive and its probability of waking from nest-mate stimulation. We also included the number of stimulations each ant received before becoming active as a fixed effect in the GLMMs to assess whether 'response thresholds' could better explain the activation patterns of individuals. The idea of response thresholds (where workers perform an action only after their perception of a stimulus exceeds an internal threshold) is commonly used to explain division of labour and other aspects of collective behaviour in social insects [34–36]. If a basic kind of response threshold system was at play here, it would mean that an inactive ant's probability of activation would depend on the cumulative number of physical stimulations she receives after becoming inactive. Additionally, we estimated stimulation survival curves relating the probability of an inactive ant ignoring a stimulation event with how long that ant had been inactive.

Although this analysis may provide evidence for differences between the two species in how workers respond to physical contact, the workers were selected for survival analysis based on if they had become active during a *single* colony cycle. This sample may therefore underestimate the variation in refractory periods exhibited by workers in both species. To investigate the range of possible refractory periods in colonies, we also monitored individuals using an additional method. We selected four colonies of each species and randomly (i.e. haphazardly) chose five ants every 30 min of a colony's recording over 9 h (resulting in 45 observations per colony), identified the time when each ant became inactive closest to these 30 min intervals and recorded the duration that each ant spent inactive before either activating spontaneously or through stimulation. We used this set of inactivity durations as a proxy to estimate the range of refractory periods possible in each species. Finally, we also manually gathered data on the typical amount of time workers spend active when they are inside their nests. To do this, we selected 11 focal ants from one recording of each species (colony sizes: *Leptothorax* sp. W = 18; *L. crassipilis* = 31). For each focal ant, we recorded all physical interactions as outlined above along with every time the ant became either active or inactive for 3 h or until the focal ant left the nest to forage.

## 2.5. Model description and simulations

We built a model of collective ant activity cycles by first considering the two known processes that cause an ant to become active: (i) spontaneous activation and (ii) nest-mate stimulation. We combined these processes into a simple algorithm followed by individual ants (figure 5a). Individual ants could be in two possible states: active or inactive. When an ant becomes active, it remains so for a fixed duration ($A$). While active, the ant will roam in a random walk through the simulation arena, where it can potentially awaken inactive ants it encounters (nest-mate stimulation). While active, walking ants randomly pick a heading within 45° of their current orientation and move one step in that direction. The two-dimensional arena

(grid) that simulated ants could explore was bounded, and if an ant reached an edge, it would select an integer in the range [0, 180), rotate by that many degrees and continue moving.

We simulated our model in the NETLOGO language [6] using aggregates on a grid whose size was held constant at $32 \times 32$ patches (each patch is a square of $1 \times 1$ arbitrary units of length). Individual agents (ants) could move on the grid (i.e. between patches) while in their active state. A stimulation event was defined as the moment an active and inactive agent became at least 1 length unit apart. Ants were allowed to freely pass through one another (i.e. more than one ant could occupy the same patch). If two inactive agents occupy the same patch and one of them becomes active, this would therefore also qualify as a stimulation event if the ants were within 1 length unit of each other. The random walk of simulated active ants (moving 1 unit every time step in a direction $\pm$ [0, 45) degrees of its current heading) is similar to other models of random ant movement [7]. The relative amount a simulated ant moves in each time step is approximately equal to 1 s of movement in real ants. Although the precise walking and interaction patterns of *Leptothorax* are not directly relevant to the research questions we are addressing with our model, we also ran simulations where the amount of stochasticity in the random walk of agents was varied to see if this had any impact on our model's results. This was accomplished by having agents determine the direction of their next step in the arena by adding either an integer in the range $\pm$ [0, 5) degrees to their current heading or adding an integer from the range $\pm$ [0, 360) degrees to their current heading. Agents in simulations where headings were adjusted by $\pm$ [0, 360) degrees at each walking step thus had fully random walks, and agents in simulations that adjusted headings by only $\pm$ [0, 5) degrees had straighter and more predictable walking paths.

Every time an ant becomes inactive, two parameters are set: (i) the length of time the ant will remain inactive before activating ($S$; i.e. spontaneous activation) and (ii) the length of time the ant will ignore contacts from other ants ($R$; i.e. refractory period). These parameters are set by sampling from predefined distributions of intrinsic inactivity durations and stimulation refractory periods, respectively. The level of noise (uncertainty) in individual ant behaviour can be controlled by modifying the two underlying distributions from which parameters $R$ and $S$ are sampled. All simulations were run using a colony size of 50 ants, and all simulations consisted of 100 001 time steps (corresponding to roughly 27.8 h of live ant observation). Although this study was not designed to assess the effect of worker density, the choice of using 50 ants in simulated colonies results in a biologically reasonable population density. Because worker *Leptothorax* ants are approximately 3 mm long and agents in the model are essentially 1 unit/patch long, the size of a patch in the model can be thought of as being approximately $3 \times 3$ mm. The area of the simulated nests is thus approximately $(32 \times 3)^2 = 9216 \text{ mm}^2$, and the area of the artificial circular nests from our empirical observations is $\pi \times (19)^2 = 1134 \text{ mm}^2$. Since the populations of our *Leptothorax* colonies ranged from 7 to 248 individuals, 50 simulated ants occupying approximately $9216 \text{ mm}^2$ falls near the kind of densities that the smaller colonies in our artificial nests experienced.

Using the empirical data collected from individuals to parametrize our model, we ran simulations to determine if any of the observed collective-level behaviours seen in real colonies of either species could be reproduced by the model. The mean for parameter $S$ was determined for both species by taking the average value (rounded to the nearest integer) of isolated individuals' average IBI values. Simulated ants would then set $S$ each time they became inactive by sampling from an exponential distribution with a rate parameter of $\lambda = 1/\langle S \rangle$. Parameter $R$ was determined for *L. crassipilis* by taking the mean duration of inactivity of ants inside colonies, and ants would set their $R$ when inactive by sampling from an exponential distribution with a rate parameter of $\lambda = 1/\langle R \rangle$. Parameter $R$ was instead determined for *Leptothorax* sp. W by having ants sample from a uniform distribution whose limits were the edges of the interquartile range of inactivity durations of ants inside colonies. This difference in how the parameter $R$ was sampled in our model was motivated by the difference we found between the two species in the distributions of inactivity durations from ants inside their colonies (see §3). Because the durations of activity had less variation than the durations of inactivity, we set $A$ as a constant in both species. Parameter $A$ was determined for each species using their median durations of activity when in nests with conspecifics.

The parameters for artificial *Leptothorax* sp. W colonies were as follows: $R \sim$ Uniform(530 sec, 1415 sec); $S \sim$ Exp(3824 sec) and $A = 218$ sec. The parameters for artificial *L. crassipilis* colonies were as follows: $R \sim$ Exp(1513 sec); $S \sim$ Exp(2385 sec) and $A = 138$ sec. Simulations of colonies always used aggregates with 50 ants with an initial condition of 25 ants starting in the active state and 25 ants starting in the inactive state.

To understand how the refractory period and its associated noise might modify the tempo of collective oscillations in the model and whether or not these factors can lead to multi-rhythmic

behaviour, we also conducted simulations where we systematically varied the refractive period ($R$) along with the amplitude of refractory noise ($\Omega$). Starting with a fixed value of $R$, we ran simulations where ants could sample their refractory periods from a uniform distribution with a progressively increasing width whose mean remained $R$. For example, if $\Omega = 300$ and $\langle R \rangle = 1100$ s, every time an ant becomes inactive, it will determine its refractory period by randomly selecting any integer in the range [800 s, 1400 s] with equal probability. To ensure arrhythmic spontaneous activation of individuals, the values of parameter $S$ were sampled from an exponential distribution with a rate parameter of $\lambda = 1/\langle S \rangle$.

# 3. Results

## 3.1. Activity patterns of colonies

Although both *Leptothorax* species possess STACs, we found the distributions of colony cycle periods differ significantly between them (figure 1*b*; Kolmogorov–Smirnov test: D = 0.734, $p < 0.0001$). *Leptothorax* sp. W shows little variation between colonies in the dominant period of its STACs; colonies oscillate with a period of $21.2 \pm 4.6$ min (figure 1*e*; electronic supplementary material, video S5). These period values are similar to those reported for the related species *L. acervorum* [29,32]. By contrast, *L. crassipilis* has an average period of $56.8 \pm 39.9$ min, and colonies expressed multiple oscillation periodicities ranging from 16.0 to 169.4 min (figure 1*d*). The dominant period of the collective oscillations was not correlated with colony size in either species (*L. crassipilis*, Pearson correlation: $r = 0.1009$, $p = 0.6024$; *Leptothorax* sp. W, Pearson correlation: $r = -0.0848$, $p = 0.7463$; electronic supplementary material, figure S2).

An examination of the longer, 35 h colony time series indicates a potential for multi-rhythmic collective cycles in *Leptothorax* (figure 2). In multiple colonies from both species, more than one distinct STAC periods co-occur within the same time series. This can be seen in the Lomb–Scargle periodograms of the time series as at least two clear peaks in the power spectra (figure 2*a–c*). For instance, in the *Leptothorax* sp. W colony presented in figure 2*a*, the dominant oscillation period is approximately 20 min, and this rhythm pervades throughout the 35 h recording, yet the periodogram reveals a secondary rhythm with a period of about 3 h. This longer rhythm becomes visually obvious when larger amounts of smoothing are applied to the time series (see green line of figure 2*a*). *Leptothorax crassipilis* colonies also exhibited multiple rhythms within the same time series (figure 2*b,c*). In colonies that had both a 'long' and 'short' rhythm, the long rhythm occurred simultaneously with the shorter one, but the long rhythms also give the impression that they might sometimes fade out, leaving just the faster rhythm. Not all colonies expressed multiple rhythms. The *L. crassipilis* colony L4, for example, has just one clear peak in its periodogram. This peak occurs at 2.6 h, and the time series plot shows that the long cycles persist for the entire activity record (figure 2*d*). As evidenced by the two tall peaks that emerge when the rescaled Lomb–Scargle power spectra of all 35 h time series are summed together, several of the 'long' periods from different colonies are all very close to 3.8 h, and several of the 'shorter' periods in different *L. crassipilis* colonies are all very close to 1.4 h (figure 3).

## 3.2. Activity patterns of isolated individual ants

We found the activity of isolated workers of both species showed sustained intervals of inactivity interspersed with short bursts of movement (figure 4*a,b*). Worker activity resembled trains of action potentials in spiking neurons and were accordingly analysed by calculating the mean time between activity spikes (IBI) and the CV of inter-beat times, two common metrics used in neuroscience [37]. Processions of activity spikes in workers of *Leptothorax* sp. W were largely arrhythmic (CV = $0.97 \pm 0.25$, figure 4*c*) and were often indistinguishable from a Poisson process (i.e. CV = 1). A lower CV for *L. crassipilis* spike trains (CV = $0.74 \pm 0.16$, figure 4*c*) reveals that activity bursts are more predictable in this species than in *Leptothorax* sp. W (Linear mixed-effects model (LME): $t_{20} = 3.38$ $p = 0.003$). The average interval between consecutive spikes in *L. crassipilis* individuals (IBI = $39.7 \pm 17.3$ min, figure 4*d*) is also shorter than those of *Leptothorax* sp. W (IBI = $63.7 \pm 34.6$ min, figure 4*d*) but not significantly so (LME: $t_{20} = 1.96$, $p = 0.064$). We also observed substantial intraspecific variation in CV and mean IBI values across workers of both species (figure 4*c,d*).

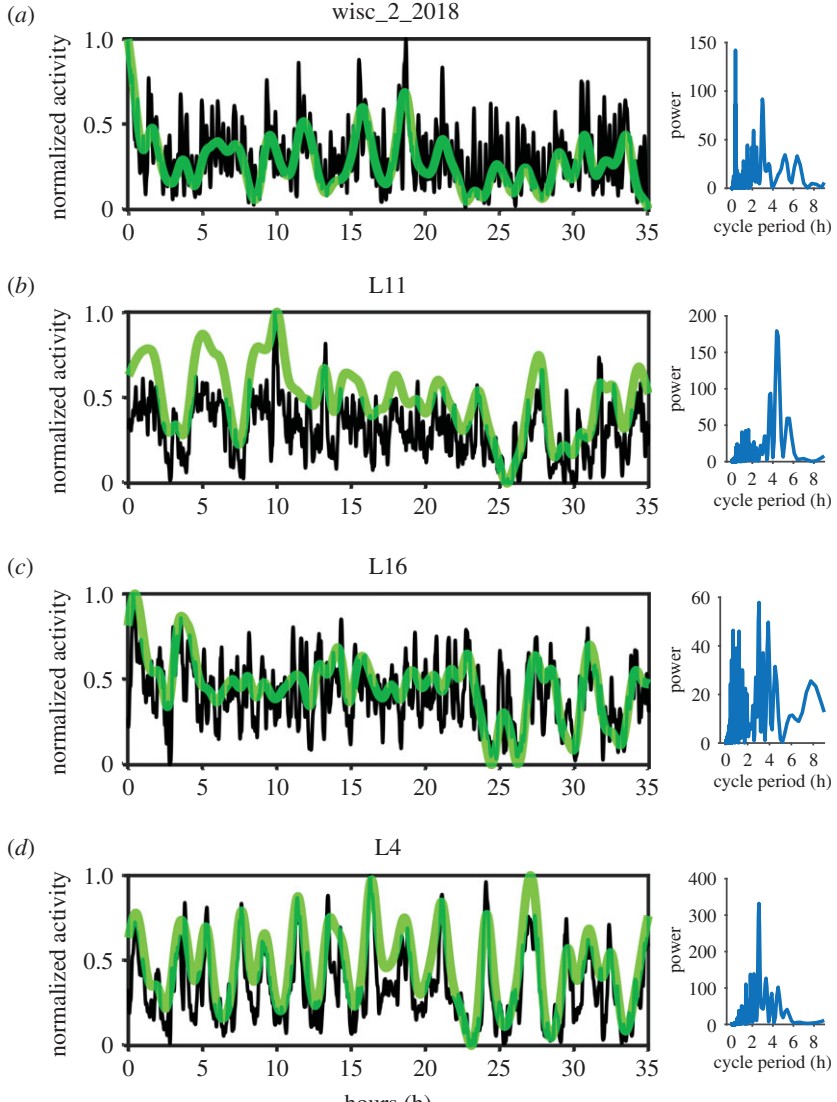

**Figure 2.** Longer time series from colonies. Example time series from 35 h recordings of a *Leptothorax* sp. W colony (*a*) and three *L. crassipilis* colonies (*b–d*). The black curves in each plot depict the collective activity of colonies after being smoothed using a Gaussian-weighted moving average with a window of 15 points. The translucent green curves depict the same time series as the black curves except with a smoothing window of 200 data points. The time series have been rescaled to fall between 0 and 1 in each panel. Lomb–Scargle periodograms are plotted to the right of their corresponding time series. The periodograms were made using the detrended time series, and no smoothing was applied to the time series prior to this particular analysis. Multiple collective oscillation frequencies can occur in the same activity record.

## 3.3. Activity propagation through individual physical contact and typical durations of activity

For both species, we found the longer a focal ant was inactive, the higher the likelihood that physical stimulation would induce activity (*Leptothorax* sp. W, GLMM: $z = 4.677$, $p < 0.0001$; *L. crassipilis*, GLMM: $z = 2.976$, $p = 0.0029$). However, the effect was significantly weaker in *L. crassipilis* than in *Leptothorax* sp. W (GLMM species/time interaction: $z = -2.941$, $p = 0.0033$). Furthermore, the number of interactions that an ant received was not significantly associated with becoming active in either species (*Leptothorax* sp. W, GLMM: $z = -1.371$, $p = 0.1703$; *L. crassipilis*, GLMM: $z = 1.155$, $p = 0.2482$). The effect of the number of interactions on activation was also not significantly different between species (GLMM species/no. of stimulations interaction: $z = -1.757$, $p = 0.0789$). This is consistent with the idea that workers have a refractory period during which they will tend not to respond to nestmate stimulation [19,29].

An inspection of the survival curves reveals that after 10 mins of inactivity there was a distinct decline in the probability that *Leptothorax* sp. W would remain inactive, possibly suggesting a less variable

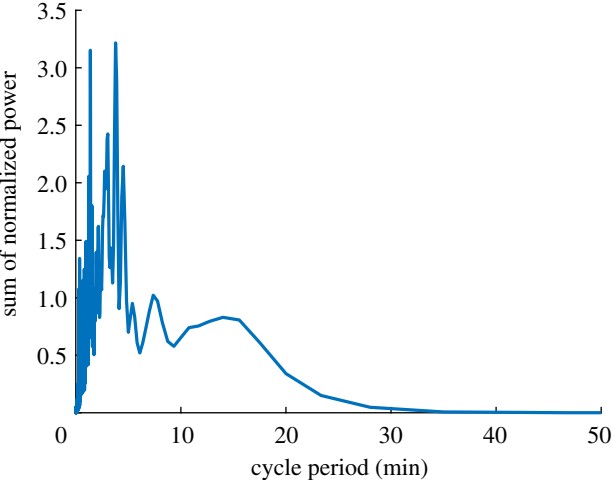

**Figure 3.** Summed power spectra from all 35 h colony time series. The periodogram depicted here was created by rescaling the Lomb–Scargle power spectra of all eight of the 35 h colony recordings and summing them together. There are two distinct peaks, which indicates that multiple colonies exhibit these particular periodicities.

refractory period than that seen in *L. crassipilis* (figure 4*e*). We also found that the probability of ignoring the stimulus decreased significantly more quickly for *Leptothorax* sp. W than *L. crassipilis* (figure 4*e*, Logrank test: $\chi^2_1 = 8.1$, $p = 0.005$).

The distributions of each species' individual ant inactivity durations are distinct (Kolmogorov–Smirnov test: D = 0.189, $p = 0.0033$, figure 4*f*). The aggregate data from *Leptothorax* sp. W are right skewed and unimodal, but the distribution of *L. crassipilis* is more consistent with an exponential distribution. Based on our observations of individuals over 3 h, the mean duration of activity inside nests is not significantly different between species (LME: $t_{18} = 1.29$, $p = 0.212$).

## 3.4. Model simulations

The appearance of rhythmic oscillations in our model occurs despite noise in individual refractory periods. Specifically, when parametrized to approximate the individual-level data from *Leptothorax* sp. W and using a uniform distribution for parameter $R$ to introduce refractory noise (see electronic supplementary material), this model generates individuals that are erratic when on their own but who can oscillate rhythmically when other ants are present. These cycles are, qualitatively, like those seen in real colonies (figure 5*b*). However, according to our wavelet analysis, the dominant cycle periods of simulated *Leptothorax* sp. W colonies (11.92 ± 3.41 min) are shorter than those seen in real colonies (figure 5*c*). Although an exponential distribution of refractory periods also generates collective oscillations, when the model's parameters are set to match *L. crassipilis*, the resulting cycles (8.07 ± 1.62 min) do *not* exhibit the large range of cycle periods seen in real colonies of this species (figure 5*c*). When the random walk of agents is modified to be less stochastic (next step is their old heading ± [0, 5) degrees), there is no impact on the dominant periods of the model's simulations when using the parameter set of either species (electronic supplementary material, figure S3). Making the agents choose the direction of their next step completely randomly (next step is their old heading ± [0, 360) degrees) also does not greatly affect the dominant periods of simulated *L. crassipilis* colonies (electronic supplementary material, figure S3). However, completely random motion does result in the simulated time series of *Leptothorax* sp. W colonies having dominant periods that more closely match those of real colonies (electronic supplementary material, figure S3).

When we examined the effect of refractory period length and refractory noise on the model's rhythmic behaviour, we noticed that the long simulation outputs had stationary means, so we used Lomb–Scargle periodograms to analyse their spectral properties and to find the period with the highest spectral peak in each time series (i.e. the dominant period). Inspection of simulation time series and their periodograms reveals that multi-rhythmicity is possible in this model (figure 6). When there is no refractory noise, the dominant collective period increases linearly with the refractory period $R$ (figure 6*a,c*). However, once $R$ exceeds a threshold value (in this case $R = 900$ s), birhythmic collective oscillations become common; simulated colonies intermittently switch between a long cycle

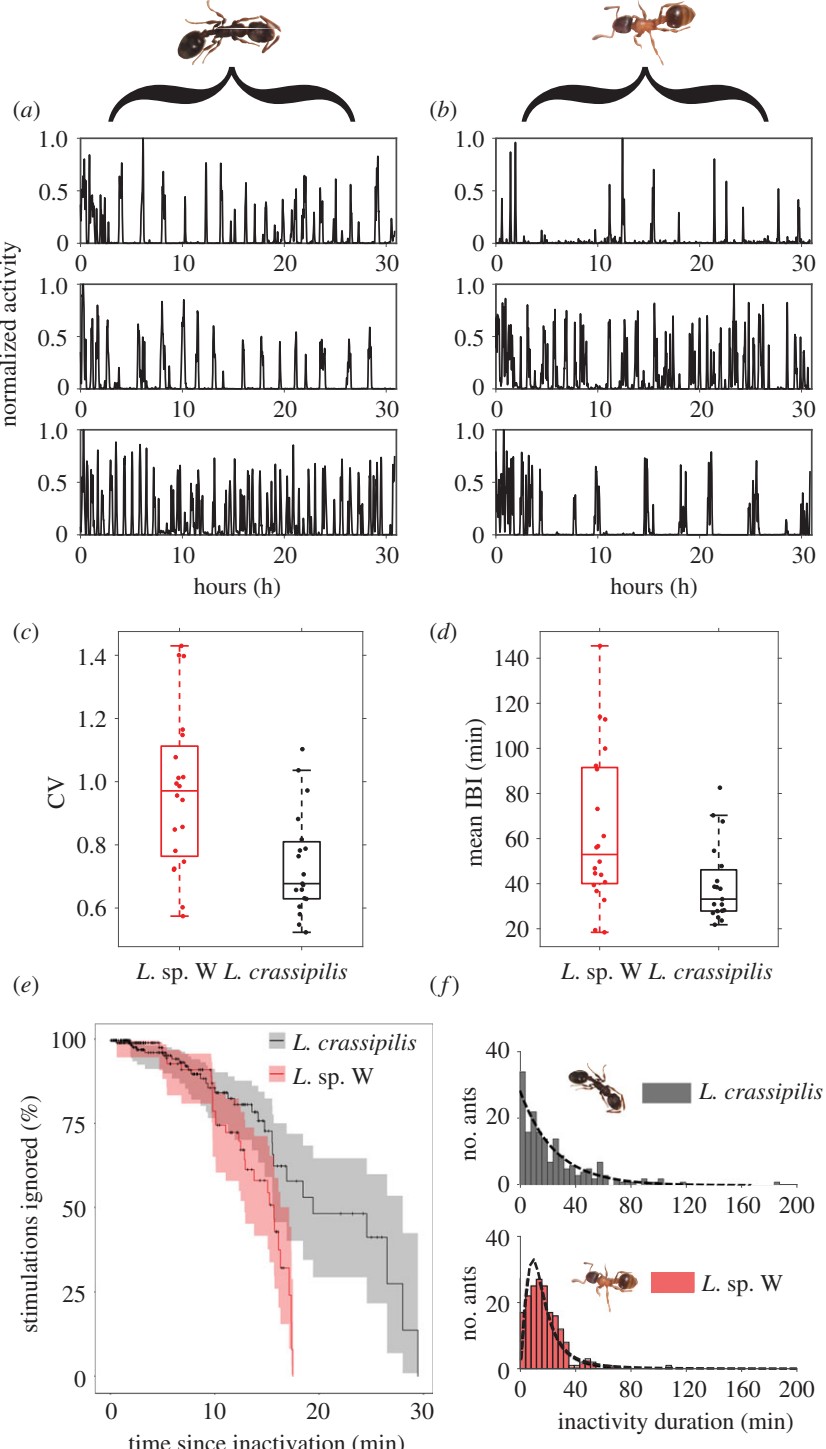

**Figure 4.** Empirical data from individuals. (*a*) Time series of locomotor activity from three representative-isolated *L. crassipilis* singletons and (*b*) three *Leptothorax* sp. W singletons. Activity time series of individuals were normalized, so that each ant's movement distances (originally in pixels) were rescaled to fall between 0 and 1. Box plots comparing (*c*) the CV and (*d*) mean IBI of activity time series from isolated singletons between species. Box plot points are horizontally offset for visibility. Isolated *Leptothorax* sp. W singletons have greater variance in CV and often have longer spike intervals than *L. crassipilis*. (*e*) Kaplan–Meier 'survival' curve estimates of the probability that an ant will ignore a stimulation as a function of the time an ant has spent inactive. Shaded areas represent log–log 95% CIs. (*f*) Differing distributions of individual ant inactivity durations between *Leptothorax* species. Data for each species is combined from observations from four colonies. The dotted lines depict the best fit exponential distribution for *L. crassipilis* and best fit log–logistic distribution for *Leptothorax* sp. W.

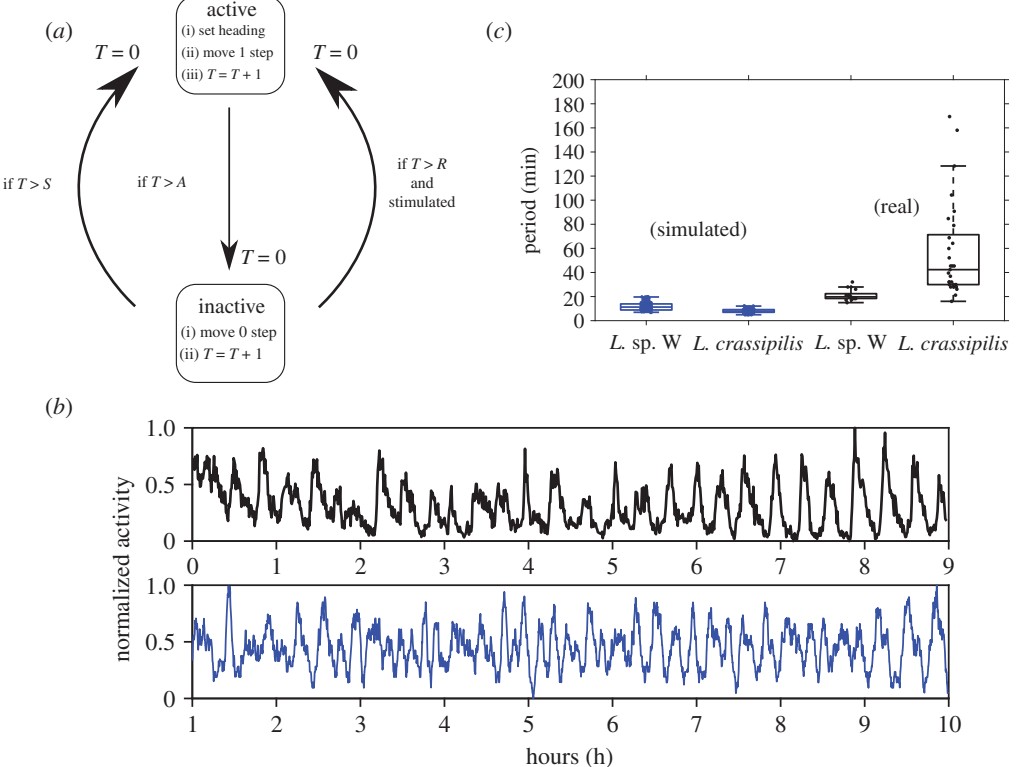

**Figure 5.** Algorithm used by modelled ants, and simulated colonies versus real colonies. (*a*) Schematic diagram of individual ant behaviour used in the model. Boxes depict the two possible states ants may occupy. Text in each box describes the sequence of behaviours executed by an ant during each time step (i.e. second) spent in the corresponding state. Arrows indicate the ways that ants may transition between the two states. Whenever an ant switches states it sets its personal clock $T$ to 0. (*b*) Plots of activity traces for an example *Leptothorax* sp. W colony (black line) and a segment of a simulated *Leptothorax* sp. W colony (blue line) over approximately 9 h (32 400 time steps of the simulation). Data were normalized to fall between 0 and 1. (*c*) Box plots showing the cycle periods of real (black) versus simulated (blue) colonies of both *Leptothorax* species. The simulated data for *L. crassipilis* and *Leptothorax* sp. W used one hundred separate simulations (100 001 time steps long) for each species.

and a short cycle (figure 6*a*,*d*). Longer collective cycles are thus more susceptible to multi-rhythmic behaviour. The addition of refractory noise has a nonlinear effect on multi-rhythmicity (figure 6*b*). Small amounts of noise (e.g. $\Omega = 50$) have no effect on the collective oscillations, but larger amounts of noise reduce the birhythmicity associated with larger values of $R$, causing simulated colonies to favour the longer cycle (figure 6*b*,*e*). Additionally, when the refractory periods of agents are determined by sampling from an exponential distribution, clear evidence for multi-rhythmicity does not appear in any of the resulting simulations at all (electronic supplementary material, figure S4).

## 4. Discussion

Our findings show that there are detectable interspecific and intraspecific differences in the activity patterns of singleton workers and whole colonies of *Leptothorax*. We also show that, in both of the studied species, multiple collective oscillation frequencies can be present in the same colony. The collective oscillations and individual-level locomotor patterns of *Leptothorax* ants are therefore more diverse than previously known. Although both of the evaluated species have collective activity cycles, the two species vary in (i) the distributions of dominant colony oscillation frequencies, (ii) the predictability of isolated worker activations, and (iii) the distributions of worker inactivity durations. Our model simulations corroborate that collective oscillations naturally manifest in ants that move spontaneously and stimulate conspecifics, even when individuals lack a fundamental underlying rhythmicity or possess noise in their refractory periods. For some parameter values, collective cycles may also exhibit switching between different frequency regimes, yet the occurrence of such multi-rhythmicity is reduced in the model when noise is added to the refractory periods of each worker.

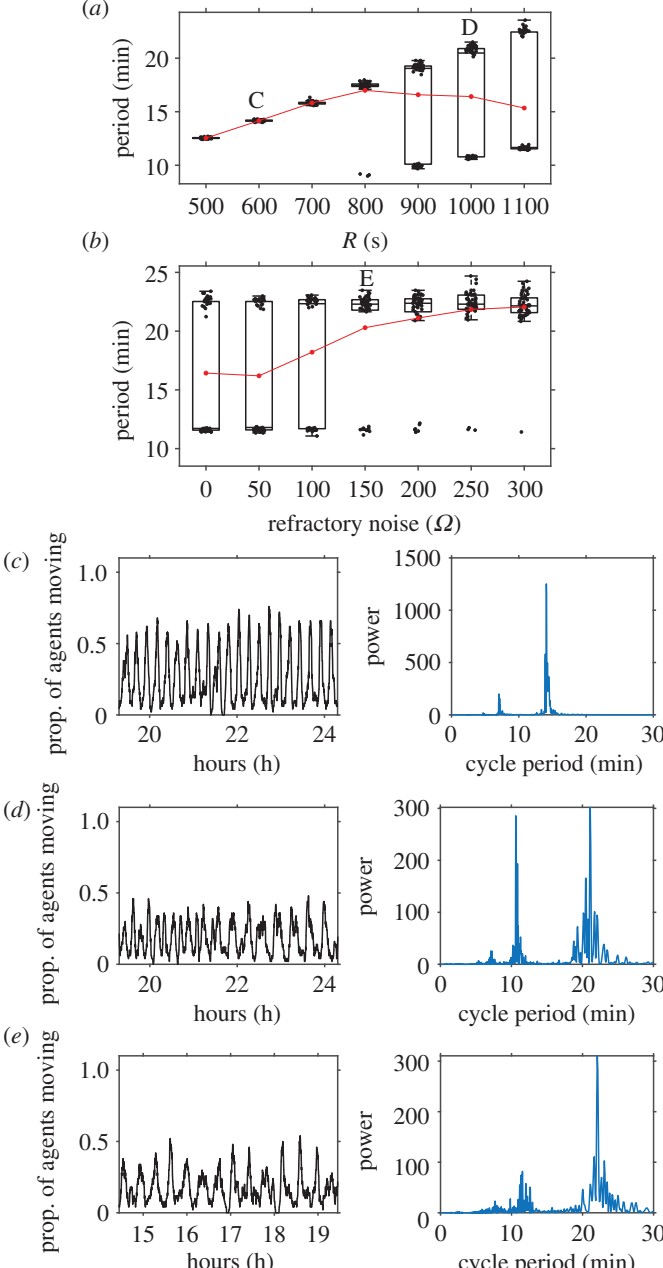

**Figure 6.** Effect of refractory period length and refractory noise on collective oscillations. (*a*) Boxplots of the dominant period (determined through Lomb–Scargle periodograms) for simulations using different values of $R$ (refractive period). The red dots indicate the mean of each box. There are 50 simulations of 100 001 time steps for each $R$ value, and the other model parameters were held constant at: $\langle S \rangle = 4050$ s, $A = 216$ s and $\Omega = 0$. (*b*) Boxplots of the dominant period for simulations using different levels of refractory noise $\Omega$. There are 50 simulations of 100 001 time steps for each $\Omega$ value. The other model parameters were held constant at: $\langle S \rangle = 4050$ s, $A = 216$ s and $\langle R \rangle = 1100$ s. (*c*–*e*) Plots of unprocessed time series segments showing the proportion of agents active over time for selected simulation runs along with the associated Lomb–Scargle periodograms of the entire time series. Letters C to E in panels (*a*) and (*b*) correspond to the respective locations of the example time series and periodograms. Increasing the refractive period of individual ants results in longer collective cycles but also encourages multi-rhythmicity. The addition of refractory noise reduces the amount of switching between collective frequencies.

In excitable media and certain network configurations of neurons and coupled oscillators, both noise and heterogeneity can have profound implications for collective behaviour including sometimes destroying or promoting precision and synchronization [5,6,28,38,39]. Behavioural heterogeneity between workers within social insect colonies has long been noted [40,41]. These differences are thought to be crucial to division of labour [35] and are positively linked with reproductive output [29]

and swift collective decision-making [42,43]. The contribution of noise to social synchronization in insects has received little attention, but behavioural noise is known to affect ant collective behaviour in other contexts, such as aiding colonies' decision-making in dynamic environments [44] and causing more accurate navigation during cooperative prey retrieval [45]. Models also suggest that colonies can maintain a near-optimal allocation of workers to different tasks even when the ability of ants to sense task demand is imperfect [46]. The data herein extend this idea by showing that collective oscillations in social insects need not be contingent on behavioural uniformity in workers. This result also matches the outcome of work with non-mobile excitable cellular automata that lack spontaneous activation, which similarly found that synchronization can persist despite stochasticity in refractory periods [47].

Our model reveals that multi-rhythmicity can arise in excitable systems if individuals are also capable of spontaneous individual activation and the stimulation refractory period is sufficiently long. This effect may contribute to the diverse collective-level frequencies of *L. crassipilis* and to the multiple co-occurring rhythms in both species, though this remains uncertain. The reduction in multi-rhythmicity associated with higher levels of noise in our model is reminiscent of work done on stochastic resonance and coherence resonance in other models of excitable systems, where limited amounts of noise emanating from a common external source improve coherence [6,48]. In our case, instead of an improvement in oscillator coherence, we detected less switching between collective rhythm frequencies. It has also recently been shown that adding independent and uncorrelated sources of noise separately to individual oscillators can still improve synchronization [49]. The refractory noise in our model was added independently to each ant and was therefore uncorrelated, not originating from a common source. Our finding thus uncovers a novel impact that uncorrelated noise can have on oscillations in excitable systems.

Multi-rhythmicity has been documented in a handful of physical systems and in models of biological oscillators such as the mammalian circadian clock [27], but additional research could uncover a wider set of conditions where the phenomenon occurs. The results of our simulations raise the possibility that other natural oscillatory systems or theoretical models with either mobile, non-identical or excitable elements (like aggregations of microorganisms [50], firefly swarms [51] or biological neuron models [38]) may harbour similar collective frequency switching behaviour under the right conditions, namely wherever there exists sufficient randomness in the intrinsic activations of individual components. Understanding the factors that can lead to and control multi-rhythmicity is an active area of research, as the phenomenon can be undesirable [52]. Evaluating the functional consequences of behavioural noise and heterogeneity on multi-rhythmicity in these types of systems could thus be an attractive direction for future study.

The simplicity of our model results in some limitations which should form the subject of future work. We do not know if the multi-rhythmicity seen in our model is caused by similar processes as those which lead to the multiple rhythms that we observed in the 35 h recordings of live colonies. Although we have shown that a single colony can possess multiple oscillation frequencies, genuine multi-rhythmicity involves switching between distinct frequencies. It is not yet clear if this happens reliably in *Leptothorax* or if the multiple periodicities must always occur at the same time. The origin of the long dominant periods in *L. crassipilis* also deserves more attention. Achieving long collective periods in this type of system cannot be trivially accomplished by lengthening the average refractory period of workers because of the spontaneous activation of workers. Either most ants will activate spontaneously before they are susceptible to stimulation by another worker (when $\langle R \rangle$ is long and $\langle S \rangle$ is short) or collective cycles will become arrhythmic (when both $\langle R \rangle$ and $\langle S \rangle$ are long). The long dominant periods we observed in *L. crassipilis* are inconsistent with earlier cellular automata models of STACs as well. These models can produce simulated colonies with long periods, but this results in every agent in the simulated colony being in a near constant state of activity, with periodic dips in the sustained universal activity [20,53]. This is not what happens in actual colonies. It is also worth considering that the ways in which we parametrized the model, calibrated the movement per time step and estimated the distributions of refractory periods were all simplifying approximations to make the model tractable, which can lead to inaccuracies.

There are likewise factors that we ignored in favour of generality, but which may be relevant to STACs. We did not consider behavioural heterogeneity between workers, and we treated the movements of workers as correlated random walks. However, *Temnothorax* workers in a single colony are known to vary in their average level of total activity when measured over more than a week [54]. Worker movement paths (and interactions with nest-mates) can also be influenced in complex non-random ways by the environment inside the nest. For example, workers from species that are closely

related to *L. crassipilis* and *Leptothorax* sp. W are known to spend more time in some regions of the nest than others, which are sometimes referred to as 'spatial fidelity zones' [55]. Interactions between individuals in a colony are further complicated by dominance hierarchies and avoidance behaviour [56–58]. Workers and gynes will sometimes alter their walking paths depending on the dominance rankings or identity of nearby individuals [56,57,59]. All of these factors could therefore have consequences for STACs. It is additionally possible that workers may be able to sense the current rhythm of the colony and modify their refractory period to avoid missing a colony cycle, leading to greater coherence. A more detailed exploration of activity patterns in individuals of both species and how colonies achieve synchrony is therefore necessary. Despite the inability of the current model to fully reproduce the intricacies of STACs, our model's primary insights still stand: mobile excitable systems can synchronize when agents have noisy refractory states, and birhythmicity can be diminished through the addition of refractory noise.

The present study does not resolve a central enigma surrounding STACs: why do they exist? No experiment has been able to demonstrate any advantage for colonies that possess STACs. Some investigators have suggested that STACs foster more efficient brood care, though others have argued that they might not have any adaptive significance at all [12]. Even if synchronized activity cycles themselves do not confer an inherent functional benefit, the ability to express different dominant cycle frequencies like *L. crassipilis* may still have fitness consequences. Of the six *Temnothorax*/*Leptothorax* species where STAC data are now documented [12,13,24], three species (*T. allardycei*, *L. acervorum* and *Leptothorax* sp. W) consistently exhibit oscillations of 15–30 min, two species (*T. albipennis* and *T. rugatulus*) exhibit slower oscillations of approx. 50 min and *L. crassipilis* is notable for its large variability in dominant frequency. Because colony tasks are believed to be completed primarily during times of high activity, the tempo of a colony's oscillations might dictate how rapidly it can respond to changing conditions outside the nest (e.g. detecting and exploiting food resources) or inside the nest (e.g. heightened levels of hunger in larvae). Testing more species will probably help resolve the question of whether activity cycles are adaptive and uncover new types of collective movement behaviours in ants, the most ecologically dominant terrestrial invertebrate on the planet.

Data accessibility. Data and relevant code for this research work are stored in GitHub: https://github.com/naviddio/Leptothorax_cycles and have been archived within the Zenodo Repository: https://zenodo.org/badge/latestdoi/287580941.

Authors' contributions. G.N.D.: conceptualization, data curation, formal analysis, investigation, methodology, project administration, software, visualization, writing—original draft, writing—review and editing; B.D.: methodology, writing—review and editing; C.L.: methodology, writing—review and editing; J.N.P.: funding acquisition, resources, writing—review and editing; L.R.P.: methodology, writing—review and editing; K.D.-V.: methodology, supervision, writing—review and editing.

All authors gave final approval for publication and agreed to be held accountable for the work performed therein.

Competing interests. We declare we have no competing interests.

Funding. This research was supported by the National Institutes of Health (grant no. GM115509 to J.N.P.).

Acknowledgements. We thank Brendan McEwan, Hannah Anderson, Tricia Skelton, Avani Pathak, Amy Kalbfleisch, Martina Bravo, Haolin Ye, Janice Yan and Greg Thung for help with ant colony maintenance. We are grateful to Richard T. Doering and James L. L. Lichtenstein for helping collect wild colonies. Matthew Prebus helped with species identification. We also thank Michael Guevara, Jonathan Dushoff, David Earn, Kirsten Sheehy and Dobromir Dotov for helpful discussions. Patrick Bennett and Stephen Pratt provided valuable feedback on the manuscript. Finally, we thank Reuven Dukas for access to his dissecting microscope.

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
