## [Peer Review File · Royal Society Open Science]

Review History

RSOS-210810.R0 (Original submission)

Review form: Reviewer 1

Is the manuscript scientifically sound in its present form?

Yes

Are the interpretations and conclusions justified by the results?

Yes

Is the language acceptable?

Yes

Do you have any ethical concerns with this paper?

No

Have you any concerns about statistical analyses in this paper?

No

Recommendation?

Major revision is needed (please make suggestions in comments)

Comments to the Author(s)

The authors studied short-term activity cycles in ant colonies and revealed the effects of intrinsic noise and interspecific differences in individual-level behavior on the cycles in a whole colony. This research topic is important for not only collective behavior in animal groups but also complex systems such as neural networks and swarm robotics. The manuscript is written well. However, I have some concerns about the simulation and the result.

Major

- It seems interesting that the simulation model has a bifurcation from one cycle to multirhythmicity based on parameters R and Ω . However, since it is only a simulation with narrow ranges of parameters (e.g., fixed or uniform distributed R , $\langle S \rangle = 4050$, $A = 216$), we do not really know the mechanism of these characteristics. Of course, an analytical model for this simulation would be outside of the scope of the current study. However, the authors should explore the model behavior in the simulations by changing the parameters and the distributions. For example, how does the bifurcation diagram shown in Fig. 4A, B in the case of R value following an exponential distribution like the case of *L. crassipilis*? If it cannot produce multirhythmicity, the mechanisms of multirhythmicity observed in *L. crassipilis* may differ from the hypothesis.

- Mobility of ants is explicitly implemented in the simulation model. However, it would not play an essential role in the colony rhythms because it is just a random walk. For example, I guess that the effect of the mobility would be the same as assuming that the probability that inactive becomes active via interactions is proportional to the number of currently active individuals without considering space. Please discuss the difference between the actual and simulated movements.

- line 109: When did you start and finish filming? I am concerned about an effect of circadian rhythms. There is a possibility that the multirhythmicity comes from the difference of activity in morning, afternoon, and night. There are no circadian rhythms in these species? Did you confirm that by 35 hours observation?

Minor

- line 63: though -> through

- line 236: were -> where

- lines, 249, 251, 264, 266, 268, 269, 297, 300: Please clarify S.D. or S.E.

- line 275: (-> :

- figures: Please unify upper/lower case of alphabets a, b, c,...

Review form: Reviewer 2

Is the manuscript scientifically sound in its present form?

Yes

Are the interpretations and conclusions justified by the results?

No

Is the language acceptable?

Yes

Do you have any ethical concerns with this paper?

No

Have you any concerns about statistical analyses in this paper?

No

Recommendation?

Major revision is needed (please make suggestions in comments)

Comments to the Author(s)

The authors examined the activity patterns of ant workers using two different species and found some differences between them, which were the distributions of colony oscillation frequencies, the distributions of worker inactivity durations and so on. The authors also developed a simulation model to explain collective oscillations using the empirical data. This paper is potentially interesting. However, before the paper is considered for publication, there are questions that must be answered.

Major concerns:

I have a concern about gaps between experimental results and simulation results in respect with active cycle periods. Please see my comments below. Some of them may be related to this issue.

I am not sure whether density of artificial ants in the simulation field is consistent with that in ant experiments. Can you clarify it for me? Linked to this, is there any possibility that durations of activity cycles depend on ant densities?

You have stated that "A stimulation event was defined as the moment an active and inactive agent became 1 patch apart." What happens if an active agent and an inactive agent occupy the same patch with each other? Does a stimulation event occur at that time?

In a supplementary file, you have stated that "The mean for parameter S was determined for both species by taking the average value of isolated individuals average IBI values". Also, according to a supplementary file, the parameter A is set as a fixed value in both species. However, time series analysis reveals that activity rhythms of isolated individuals are little bit complex and appear to depend on individuals. Moreover, CV is quite different between two species. Don't you simplify these parameter settings? Is it correct that simulation results adequately reproduce results found in figure 2a, 2b, 2c and 2d? Is there any possibility that parameter settings may be related to mustaches regarding active cycle periods between experiments and simulations?

Minor comments:

-Do single patches approximate to the body size of single ants?

-What is a definition of multirhythmic behaviour?

-Regarding figure 1D and 1E, I am not sure which figure is L sp. W's. Can you fix that?

-You have stated that "The aggregate data from l. sp. W is right skewed and unimodal, but the distribution...." Can you plot a best fit unimodal or exponential law for individual species in figure 2F?

Review form: Reviewer 3**Is the manuscript scientifically sound in its present form?**

No

Are the interpretations and conclusions justified by the results?

No

Is the language acceptable?

Yes

Do you have any ethical concerns with this paper?

No

Have you any concerns about statistical analyses in this paper?

No

Recommendation?

Reject

Comments to the Author(s)

General comments

This manuscript presents a study of the individual and collective activity patterns of two species of *Leptothorax* ants. The stated objectives of the study are to (1) add to the corpus of already existing data on the topic to better understand interspecific differences in individual and collective activity patterns, (2) investigate the existence of multirhythmicity in the activity pattern of ant colonies, and (3) develop a data-driven model to explore the role of individual-level "noise" on the collective activity output of an ant colony. The premises of this study are extremely interesting; indeed, the question of the origin of activity patterns in social groups is of great interest to any field interested in the emergent properties of collective systems. However, I think that this particular study falls short on each of its stated objectives, and while it certainly shows good potential, it also requires additional work to be complete.

For (1), the authors state that "empirical data are scarce and limited to just a handful of species", mainly ants of the genera *Temnothorax* and *Leptothorax*. They then proceed to present a study on two more species of... *Lepthothorax* ants, with similar colony sizes and organization as species in previous studies. To be clear, I am not suggesting that the data collected as part of this study is useless (they are not); but that if one of the objectives of the study was to, indeed, better understand interspecific differences, then the authors should have collected data on a wider array of species, ideally with different colony organization and across orders of magnitude in colony sizes. Simply comparing two close species does not allow for identifying much of the behavioral and ecological determinants of collective activity patterns. I suggest that the authors simply drop that stated objective from their introduction since they are not really pursuing it and the other objectives are interesting enough on their own.

For (2), the authors use the fact that there is more variability in the distribution of activity peaks in one of the studies species to suggest that there is "a potential for multirhythmic collective cycles". This is at best a far-fetched conclusion from the data. The authors did not even take the time to first check whether this increased variability was the product of a non-random process or not (for instance, using a modified version of the approach they used to do exactly that for individual workers; see detailed comment below). By the look of the period distribution in figure 1B, I would not be surprised to find that it is just the result of a random Poisson process with a minimum delay. If that was the case, then the authors could conclude that there certainly is NO multirhythmicity in their data instead of allowing doubt to linger on their conclusions. And if it is not random, then the authors could probably keep speculating about the presence of

multirhythmicity in their data but should be very clear that it is nearly impossible to formally demonstrate it given their data.

Finally, for (3), the proposed modeling study is very incomplete and does not really provide any new reliable insight into the observed activity patterns. First, the authors do not consider in their model and data analysis at least one alternative mechanism (stimulation threshold) that could explain the shape of their data. Modeling is not about finding a mechanism that matches the data reasonably well but about finding the most likely mechanism given the data and existing knowledge. Then, the authors, noticing that their model does not match their data, do not attempt to identify the possible cause for that large mismatch. An obvious first suspect (besides the alternative mechanism discussed before) is that, while the authors were careful to measure in their data several of the model's parameters, they completely ignored the interaction pattern between the agents and set the associated parameters to generic values without regards for the actual movement behavior of their ants. Yet, it is well-established that the interaction pattern between agents is critical to understanding the output of a collective dynamical system and it should have been investigated more carefully here. Finally, while the entire discussion about the role of "noise" on the collective output of the system is interesting in itself, it is not exactly new in the fields of collective behavior and complex systems (see detailed comment). It may be more novel in the specific context of understanding activity patterns in ant colonies but, because the model fails to convince that it is a good approximation of the observed data, the interest of the entire discussion is greatly reduced. What do we really learn from that model about the emergence of collective activity patterns in ants? Not much because we cannot really trust it is a good representation of the system in the first place.

In conclusion, I cannot recommend the publication of this manuscript at this stage. The authors have done a great deal of work collecting valuable and interesting data but there remains a significant amount of work to finish its analysis and turn it into a reliable model that can generate new insights into the hows and whys of activity patterns in social systems. The collected data have a lot of potential for interesting findings, and I hope the authors will take the time to exploit it fully.

Detailed comments

l. 60: "there is no external signal that synchronizes colonies" → "there is no evidence of external signal that synchronizes colonies"

l. 72: I would replace "noise" by "(behavioral) variability" throughout the text. "Noise" is the result of an observer's imperfect ability to measure a signal in the data and it is not a characteristic of a system (it's a characteristic of the relationship between the system and the observer). "Variability" (i.e., the fact that all parts of a system don't all behave similarly to each other or to themselves over time), on the other hand, is a characteristic of a system and can affect its global outcome ("noise" only affects the observer's perception of the outcome). These concepts are often confused because statistical variance is a combination of the variability of the system (what the observer truly wants to know) and of measurement errors.

l. 145: What is the rationale for choosing this moving average window size? Have you performed a sensitivity analysis on the window size to see how its choice affects the final results? If not, then it should be done.

l. 156: was the dominant oscillation period measured using the raw or the filtered data?

l. 178: Replace "haphazardly" with "randomly" throughout the text. While the meaning is clear, "haphazardly" also connotes a lack of care or method.

l. 182 and onward: This analysis assumes the existence of a refractory period. An alternative model would be that the ants "accumulate" stimulations until it reaches a threshold that triggers activation. Such a threshold model would also give the impression of a refractory period. You can distinguish between both models by looking at (1) whether the probability of activation increases with time since inactivation after controlling for the quantity of social stimulation, and (2) vice-versa, whether the probability of activation increases with the quantity of social stimulation after controlling for the time since inactivation. Both models can in principle explain your data, yet they assume very different underlying physiological mechanisms. Note also that threshold models are commonly used to explain (rightly or wrongly) many aspects of the division of labor in social insects, therefore should be considered in the context of this study.

l. 207-208: "When an ant becomes active, it remains so for a fixed duration". Explain why you chose to do this. It is currently buried in the supplementary material. All modeling decisions should be clearly and completely justified in the main text.

l. 218-220: "The random walk of simulated active ants [...] is similar to other models of random ant movement". This is a terrible justification that assumes essentially that all ants walk the same way as your ants. Besides, you have all the video data that you need to estimate the actual movement patterns of your ants. Even assuming a simple correlated random walk, you should be able to easily estimate the diffusion coefficient of your ants from tracking (even roughly) a few minutes of your video footage. You can then fit the movement parameters of your model to reproduce that diffusion coefficient. Given that your model relies on correctly reproducing the interaction rate between active and inactive ants, this step is absolutely critical.

l. 231-232: "Using the empirical data collected from individuals to parametrize our model (supplementary material)". How the parameterization is achieved is a critical part of the modeling process and should not be relegated to the supplementary material.

l. 254: "indicating a potential for multirhythmic collective cycles". This is a very far-fetched conclusion. Multirhythmicity is characterized by the coexistence of two or more distinct oscillatory states and can be very difficult to formally demonstrate, especially with the small number of observed bursts (relative to the length of each series that is) and the non-stationarity of the time series. You could be simply witnessing irregularly spaced bursts of activity. You can test this formally more easily using the same approach that you used for individual ants (however, you may first want to "normalize" the IBI by subtracting the value of the smallest IBI from all IBIs to account for the "cooling down" time after each peak that constrains the minimum possible IBI that the system can create).

l. 273-onwards: See my comment above. These analyses should be done while controlling for the amount of social stimulation received by the ants to account for an alternative model in which an ant activation depends on the amount of social stimulation it receives and not on the existence of a refractory period (and vice versa).

l. 320: "patters" → "patterns"

l. 335-336: "However, to our knowledge, our work is the first spatially explicit formulation of a model that combines agents that are mobile, refractory, and which activate stochastically". I am a bit confused by this statement. There are plenty of examples of models with these properties (e.g., see the firefly model in the Netlogo software that the authors used in this study). There are also several swarm robotics implementations of such models (again, following the firefly example). Are you saying that this is the first time that this type of model was used in the specific context of ant activity cycles? Even then, the old cellular automata model from Solé in the early 1990s

(reference #20 in the manuscript) has all these properties (the stochastic refractory period is implicit in their self-activation function). I would either moderate this claim or make it clearer what is truly new in the proposed model because I do not really see it.

l. 342: "The contribution of noise to the collective behaviour of social insects has received less attention than heterogeneity". Again a very confusing statement. Studying the role of heterogeneity in collective behavior is relatively recent actually (15-20 years old) but studying the role of "noise" in collective behavior has been pretty much constitutive of all modeling work since the early 1980s at least (one of my favorite papers on the topic is Deneubourg J-L, Pasteels JM, Verhaeghe JC. Probabilistic behaviour in ants: A strategy of errors? *J Theor Biol.* 1983;105: 259–271. doi:10.1016/S0022-5193(83)80007-1). Please reformulate this sentence to make it clear why you think "noise" is understudied in the field of collective behavior.

l. 376: "The simplicity of our model results in some limitations which should form the subject of future work". The main issue in my opinion is that your model does not attempt to reproduce the interaction pattern between the ants. That alone may be enough to explain the discrepancies between the model and the experiments.

l.394: "The present study does not resolve a central enigma surrounding STACs: why do they exist?" If this is a central enigma, why not testing it with your model? After all, this is what a model is for: testing hypotheses that are difficult (even impossible) to study with traditional scientific approaches. This feels like a missed opportunity to actually do something fundamentally different from previous work on the topic.

Figure 1D: Please, remove the blue (slow) and green (fast) segment indicators. The data does not allow you to conclude that there is multirhythmicity in your data and these indicators are misleading the reader as they are the product of a personal belief and not of an unbiased method.

Decision letter (RSOS-210810.R0)

Dear Dr Doering

The Editors assigned to your paper RSOS-210810 "Noise resistant synchronization and collective rhythm switching in a model of animal group locomotion" have made a decision based on their reading of the paper and any comments received from reviewers.

Regrettably, in view of the reports received, the manuscript has been rejected in its current form. However, a new manuscript may be submitted which takes into consideration these comments.

We invite you to respond to the comments supplied below and prepare a resubmission of your manuscript. Below the referees' and Editors' comments (where applicable) we provide additional requirements. We provide guidance below to help you prepare your revision.

Please note that resubmitting your manuscript does not guarantee eventual acceptance, and we do not generally allow multiple rounds of revision and resubmission, so we urge you to make every effort to fully address all of the comments at this stage. If deemed necessary by the Editors,

your manuscript will be sent back to one or more of the original reviewers for assessment. If the original reviewers are not available, we may invite new reviewers.

Please resubmit your revised manuscript and required files (see below) no later than 07-Dec-2021. Note: the ScholarOne system will 'lock' if resubmission is attempted on or after this deadline. If you do not think you will be able to meet this deadline, please contact the editorial office immediately.

Please note article processing charges apply to papers accepted for publication in Royal Society Open Science (<https://royalsocietypublishing.org/rsos/charges>). Charges will also apply to papers transferred to the journal from other Royal Society Publishing journals, as well as papers submitted as part of our collaboration with the Royal Society of Chemistry (<https://royalsocietypublishing.org/rsos/chemistry>). Fee waivers are available but must be requested when you submit your manuscript (<https://royalsocietypublishing.org/rsos/waivers>).

Thank you for submitting your manuscript to Royal Society Open Science and we look forward to receiving your resubmission. If you have any questions at all, please do not hesitate to get in touch.

on behalf of Professor Roland Bouffanais (Associate Editor) and Kevin Padian (Subject Editor)
openscience@royalsociety.org

Subject Editor Comments to Author (Professor Kevin Padian):
Comments to the Author:

Thank you for your submission. As you will see, the reviewers generally liked the plan of the study but had concerns about its execution. Because these are different and extensive among reviewers, it seems preferable to give you the opportunity to revise as needed, which may result in a substantially different manuscript, which we would consider as a new submission.

Additionally there were some concerns about the data: "I believe that all the data is in the repository. However, it could be better organized to facilitate its reanalysis. For instance, all the time series are in separate files, without time indication or information about the replicate conditions. These have to be reconstructed from other documents (main text, supp mat.). Ideally, all data from a given experiment should be grouped into a single file in tidy format (one column per variable/factor, one row per observation) accompanied by a single text file containing instructions on how to interpret the content of the data file. In addition, while some of the statistical analysis code is provided by the authors, I could not find any of the code that allowed them to transform the raw data for analysis."

I hope that these are useful in your reworking of your study, and best wishes in your revisions.

Associate Editor Comments to Author (Professor Roland Bouffanais):
Comments to the Author:

Three reviewers had the chance to assess your manuscript. Although they recognize the scientific merit of the study, they also point out to major issues with the both the experimental studies as well as the simulations. I recommend that you consider a major revision of your manuscript addressing all concerns raised.

Reviewer comments to Author:

Reviewer: 1

Comments to the Author(s)

The authors studied short-term activity cycles in ant colonies and revealed the effects of intrinsic noise and interspecific differences in individual-level behavior on the cycles in a whole colony. This research topic is important for not only collective behavior in animal groups but also complex systems such as neural networks and swarm robotics. The manuscript is written well. However, I have some concerns about the simulation and the result.

Major

- It seems interesting that the simulation model has a bifurcation from one cycle to multirhythmicity based on parameters R and Ω . However, since it is only a simulation with narrow ranges of parameters (e.g., fixed or uniform distributed R , $\mu = 4050$, $A = 216$), we do not really know the mechanism of these characteristics. Of course, an analytical model for this simulation would be outside of the scope of the current study. However, the authors should explore the model behavior in the simulations by changing the parameters and the distributions. For example, how does the bifurcation diagram shown in Fig. 4A, B in the case of R value following an exponential distribution like the case of *L. crassipilis*? If it cannot produce multirhythmicity, the mechanisms of multirhythmicity observed in *L. crassipilis* may differ from the hypothesis.

- Mobility of ants is explicitly implemented in the simulation model. However, it would not play an essential role in the colony rhythms because it is just a random walk. For example, I guess that the effect of the mobility would be the same as assuming that the probability that inactive becomes active via interactions is proportional to the number of currently active individuals without considering space. Please discuss the difference between the actual and simulated movements.

- line 109: When did you start and finish filming? I am concerned about an effect of circadian rhythms. There is a possibility that the multirhythmicity comes from the difference of activity in morning, afternoon, and night. There are no circadian rhythms in these species? Did you confirm that by 35 hours observation?

Minor

- line 63: though -> through

- line 236: were -> where

- lines, 249, 251, 264, 266, 268, 269, 297, 300: Please clarify S.D. or S.E.

- line 275: (-> :

- figures: Please unify upper/lower case of alphabets a, b, c,...

Reviewer: 2

Comments to the Author(s)

The authors examined the activity patterns of ant workers using two different species and found some differences between them, which were the distributions of colony oscillation frequencies, the distributions of worker inactivity durations and so on. The authors also developed a simulation model to explain collective oscillations using the empirical data. This paper is potentially interesting. However, before the paper is considered for publication, there are questions that must be answered.

Major concerns:

I have a concern about gaps between experimental results and simulation results in respect with active cycle periods. Please see my comments below. Some of them may be related to this issue.

I am not sure whether density of artificial ants in the simulation field is consistent with that in ant experiments. Can you clarify it for me? Linked to this, is there any possibility that durations of activity cycles depend on ant densities?

You have stated that "A stimulation event was defined as the moment an active and inactive agent became 1 patch apart." What happens if an active agent and an inactive agent occupy the same patch with each other? Does a stimulation event occur at that time?

In a supplementary file, you have stated that "The mean for parameter S was determined for both species by taking the average value of isolated individuals average IBI values". Also, according to a supplementary file, the parameter A is set as a fixed value in both species. However, time series analysis reveals that activity rhythms of isolated individuals are little bit complex and appear to depend on individuals. Moreover, CV is quite different between two species. Don't you simplify these parameter settings? Is it correct that simulation results adequately reproduce results found in figure 2a, 2b, 2c and 2d? Is there any possibility that parameter settings may be related to mustaches regarding active cycle periods between experiments and simulations?

Minor comments:

- Do single patches approximate to the body size of single ants?
- What is a definition of multirhythmic behaviour?
- Regarding figure 1D and 1E, I am not sure which figure is L sp. W's. Can you fix that?
- You have stated that "The aggregate data from l. sp. W is right skewed and unimodal, but the distribution...." Can you plot a best fit unimodal or exponential law for individual species in figure 2F?

Reviewer: 3

Comments to the Author(s)

General comments

This manuscript presents a study of the individual and collective activity patterns of two species of *Leptothorax* ants. The stated objectives of the study are to (1) add to the corpus of already existing data on the topic to better understand interspecific differences in individual and collective activity patterns, (2) investigate the existence of multirhythmicity in the activity pattern of ant colonies, and (3) develop a data-driven model to explore the role of individual-level "noise" on the collective activity output of an ant colony. The premises of this study are extremely interesting; indeed, the question of the origin of activity patterns in social groups is of great interest to any field interested in the emergent properties of collective systems. However, I think that this particular study falls short on each of its stated objectives, and while it certainly shows good potential, it also requires additional work to be complete.

For (1), the authors state that "empirical data are scarce and limited to just a handful of species", mainly ants of the genera *Temnothorax* and *Leptothorax*. They then proceed to present a study on two more species of... *Lepthothorax* ants, with similar colony sizes and organization as species in previous studies. To be clear, I am not suggesting that the data collected as part of this study is useless (they are not); but that if one of the objectives of the study was to, indeed, better understand interspecific differences, then the authors should have collected data on a wider array of species, ideally with different colony organization and across orders of magnitude in colony sizes. Simply comparing two close species does not allow for identifying much of the behavioral and ecological determinants of collective activity patterns. I suggest that the authors simply drop

that stated objective from their introduction since they are not really pursuing it and the other objectives are interesting enough on their own.

For (2), the authors use the fact that there is more variability in the distribution of activity peaks in one of the studies species to suggest that there is "a potential for multirhythmic collective cycles". This is at best a far-fetched conclusion from the data. The authors did not even take the time to first check whether this increased variability was the product of a non-random process or not (for instance, using a modified version of the approach they used to do exactly that for individual workers; see detailed comment below). By the look of the period distribution in figure 1B, I would not be surprised to find that it is just the result of a random Poisson process with a minimum delay. If that was the case, then the authors could conclude that there certainly is NO multirhythmicity in their data instead of allowing doubt to linger on their conclusions. And if it is not random, then the authors could probably keep speculating about the presence of multirhythmicity in their data but should be very clear that it is nearly impossible to formally demonstrate it given their data.

Finally, for (3), the proposed modeling study is very incomplete and does not really provide any new reliable insight into the observed activity patterns. First, the authors do not consider in their model and data analysis at least one alternative mechanism (stimulation threshold) that could explain the shape of their data. Modeling is not about finding a mechanism that matches the data reasonably well but about finding the most likely mechanism given the data and existing knowledge. Then, the authors, noticing that their model does not match their data, do not attempt to identify the possible cause for that large mismatch. An obvious first suspect (besides the alternative mechanism discussed before) is that, while the authors were careful to measure in their data several of the model's parameters, they completely ignored the interaction pattern between the agents and set the associated parameters to generic values without regards for the actual movement behavior of their ants. Yet, it is well-established that the interaction pattern between agents is critical to understanding the output of a collective dynamical system and it should have been investigated more carefully here. Finally, while the entire discussion about the role of "noise" on the collective output of the system is interesting in itself, it is not exactly new in the fields of collective behavior and complex systems (see detailed comment). It may be more novel in the specific context of understanding activity patterns in ant colonies but, because the model fails to convince that it is a good approximation of the observed data, the interest of the entire discussion is greatly reduced. What do we really learn from that model about the emergence of collective activity patterns in ants? Not much because we cannot really trust it is a good representation of the system in the first place.

In conclusion, I cannot recommend the publication of this manuscript at this stage. The authors have done a great deal of work collecting valuable and interesting data but there remains a significant amount of work to finish its analysis and turn it into a reliable model that can generate new insights into the hows and whys of activity patterns in social systems. The collected data have a lot of potential for interesting findings, and I hope the authors will take the time to exploit it fully.

Detailed comments

l. 60: "there is no external signal that synchronizes colonies" → "there is no evidence of external signal that synchronizes colonies"

l. 72: I would replace "noise" by "(behavioral) variability" throughout the text. "Noise" is the result of an observer's imperfect ability to measure a signal in the data and it is not a characteristic of a system (it's a characteristic of the relationship between the system and the observer). "Variability" (i.e., the fact that all parts of a system don't all behave similarly to each other or to themselves

over time), on the other hand, is a characteristic of a system and can affect its global outcome ("noise" only affects the observer's perception of the outcome). These concepts are often confused because statistical variance is a combination of the variability of the system (what the observer truly wants to know) and of measurement errors.

l. 145: What is the rationale for choosing this moving average window size? Have you performed a sensitivity analysis on the window size to see how its choice affects the final results? If not, then it should be done.

l. 156: was the dominant oscillation period measured using the raw or the filtered data?

l. 178: Replace "haphazardly" with "randomly" throughout the text. While the meaning is clear, "haphazardly" also connotes a lack of care or method.

l. 182 and onward: This analysis assumes the existence of a refractory period. An alternative model would be that the ants "accumulate" stimulations until it reaches a threshold that triggers activation. Such a threshold model would also give the impression of a refractory period. You can distinguish between both models by looking at (1) whether the probability of activation increases with time since inactivation after controlling for the quantity of social stimulation, and (2) vice-versa, whether the probability of activation increases with the quantity of social stimulation after controlling for the time since inactivation. Both models can in principle explain your data, yet they assume very different underlying physiological mechanisms. Note also that threshold models are commonly used to explain (rightly or wrongly) many aspects of the division of labor in social insects, therefore should be considered in the context of this study.

l. 207-208: "When an ant becomes active, it remains so for a fixed duration". Explain why you chose to do this. It is currently buried in the supplementary material. All modeling decisions should be clearly and completely justified in the main text.

l. 218-220: "The random walk of simulated active ants [...] is similar to other models of random ant movement". This is a terrible justification that assumes essentially that all ants walk the same way as your ants. Besides, you have all the video data that you need to estimate the actual movement patterns of your ants. Even assuming a simple correlated random walk, you should be able to easily estimate the diffusion coefficient of your ants from tracking (even roughly) a few minutes of your video footage. You can then fit the movement parameters of your model to reproduce that diffusion coefficient. Given that your model relies on correctly reproducing the interaction rate between active and inactive ants, this step is absolutely critical.

l. 231-232: "Using the empirical data collected from individuals to parametrize our model (supplementary material)". How the parameterization is achieved is a critical part of the modeling process and should not be relegated to the supplementary material.

l. 254: "indicating a potential for multirhythmic collective cycles". This is a very far-fetched conclusion. Multirhythmicity is characterized by the coexistence of two or more distinct oscillatory states and can be very difficult to formally demonstrate, especially with the small number of observed bursts (relative to the length of each series that is) and the non-stationarity of the time series. You could be simply witnessing irregularly spaced bursts of activity. You can test this formally more easily using the same approach that you used for individual ants (however, you may first want to "normalize" the IBI by subtracting the value of the smallest IBI from all IBIs to account for the "cooling down" time after each peak that constrains the minimum possible IBI that the system can create).

l. 273-onwards: See my comment above. These analyses should be done while controlling for the amount of social stimulation received by the ants to account for an alternative model in which an ant activation depends on the amount of social stimulation it receives and not on the existence of a refractory period (and vice versa).

l. 320: "patters" → "patterns"

l. 335-336: "However, to our knowledge, our work is the first spatially explicit formulation of a model that combines agents that are mobile, refractory, and which activate stochastically". I am a bit confused by this statement. There are plenty of examples of models with these properties (e.g., see the firefly model in the Netlogo software that the authors used in this study). There are also several swarm robotics implementations of such models (again, following the firefly example). Are you saying that this is the first time that this type of model was used in the specific context of ant activity cycles? Even then, the old cellular automata model from Solé in the early 1990s (reference #20 in the manuscript) has all these properties (the stochastic refractory period is implicit in their self-activation function). I would either moderate this claim or make it clearer what is truly new in the proposed model because I do not really see it.

l. 342: "The contribution of noise to the collective behaviour of social insects has received less attention than heterogeneity". Again a very confusing statement. Studying the role of heterogeneity in collective behavior is relatively recent actually (15-20 years old) but studying the role of "noise" in collective behavior has been pretty much constitutive of all modeling work since the early 1980s at least (one of my favorite papers on the topic is Deneubourg J-L, Pasteels JM, Verhaeghe JC. Probabilistic behaviour in ants: A strategy of errors? *J Theor Biol.* 1983;105: 259–271. doi:10.1016/S0022-5193(83)80007-1). Please reformulate this sentence to make it clear why you think "noise" is understudied in the field of collective behavior.

l. 376: "The simplicity of our model results in some limitations which should form the subject of future work". The main issue in my opinion is that your model does not attempt to reproduce the interaction pattern between the ants. That alone may be enough to explain the discrepancies between the model and the experiments.

l.394: "The present study does not resolve a central enigma surrounding STACs: why do they exist?" If this is a central enigma, why not testing it with your model? After all, this is what a model is for: testing hypotheses that are difficult (even impossible) to study with traditional scientific approaches. This feels like a missed opportunity to actually do something fundamentally different from previous work on the topic.

Figure 1D: Please, remove the blue (slow) and green (fast) segment indicators. The data does not allow you to conclude that there is multirhythmicity in your data and these indicators are misleading the reader as they are the product of a personal belief and not of an unbiased method.

===PREPARING YOUR MANUSCRIPT===

Your revised paper should include the changes requested by the referees and Editors of your manuscript. You should provide two versions of this manuscript and both versions must be provided in an editable format:
 one version identifying all the changes that have been made (for instance, in coloured highlight, in bold text, or tracked changes);
 a 'clean' version of the new manuscript that incorporates the changes made, but does not highlight them. This version will be used for typesetting if your manuscript is accepted.

===PREPARING YOUR REVISION IN SCHOLARONE===

- If you are providing image files for potential cover images, please upload these at this step, and inform the editorial office you have done so. You must hold the copyright to any image provided.
- A copy of your point-by-point response to referees and Editors. This will expedite the preparation of your proof.

- Ensure that your data access statement meets the requirements at <https://royalsociety.org/journals/authors/author-guidelines/#data>. You should ensure that you cite the dataset in your reference list. If you have deposited data etc in the Dryad repository, please include both the 'For publication' link and 'For review' link at this stage.
- If you are requesting an article processing charge waiver, you must select the relevant waiver option (if requesting a discretionary waiver, the form should have been uploaded at Step 3 'File upload' above).
- If you have uploaded ESM files, please ensure you follow the guidance at <https://royalsociety.org/journals/authors/author-guidelines/#supplementary-material> to include a suitable title and informative caption. An example of appropriate titling and captioning may be found at [https://figshare.com/articles/Table_S2_from_Is_there_a_trade-off_between_peak_performance_and_performance_breadth_across_temperatures_for_aerobic_sc](https://figshare.com/articles/Table_S2_from_Is_there_a_trade-off_between_peak_performance_and_performance_breadth_across_temperatures_for_aerobic_scope_in_teleost_fishes_/3843624) ope_in_teleost_fishes_/3843624.

Author's Response to Decision Letter for (RSOS-210810.R0)

See Appendix A.

RSOS-211908.R0

Review form: Reviewer 1

Is the manuscript scientifically sound in its present form?

Yes

Are the interpretations and conclusions justified by the results?

Yes

Is the language acceptable?

Yes

Do you have any ethical concerns with this paper?

No

Have you any concerns about statistical analyses in this paper?

No

Recommendation?

Accept as is

Comments to the Author(s)

The authors have done a great job in addressing my concerns in the revision of the manuscript. Of course, there are some limitations of this study. But, it's been discussed enough.

Review form: Reviewer 2**Is the manuscript scientifically sound in its present form?**

No

Are the interpretations and conclusions justified by the results?

No

Is the language acceptable?

Yes

Do you have any ethical concerns with this paper?

No

Have you any concerns about statistical analyses in this paper?

No

Recommendation?

Reject

Comments to the Author(s)

The authors have revised the article. However, I do not think that it is yet ready for publication.

Firstly, I have a still concern about gaps between experimental results and simulation results in respect with active cycle periods. Although the authors have stated that "(L384) However, making the agents choose the direction of their next step completely randomly...", this is not applied to the case of *L. crassipilis*. Furthermore, it is not likely that ants choose the direction of their next step randomly.

Secondly, the parameter R following an exponential distribution seems to fail to produce multirhythmicity, which is conflict with the experimental fact. Maybe, the mechanism of multirhythmicity observed in *L. crassipilis* is different from the model. The authors have stated that "(L493) ...our model's primary insights still stand: mobile excitable systems can synchronize when agents have noisy refractory states, and birhythmicity can be diminished through the addition of refractory noise." I agree with that. At the same time however, I get impression that the authors try replicating experimental phenomena using proposed model. In such circumstances, I cannot ignore these inconsistencies between simulations and experiments.

As it is already a revised version of the article, I am sorry but I cannot recommend acceptance or minor revisions and therefore I rather suggest to reject the article from publication.

Review form: Reviewer 3

Is the manuscript scientifically sound in its present form?

Yes

Are the interpretations and conclusions justified by the results?

Yes

Is the language acceptable?

Yes

Do you have any ethical concerns with this paper?

No

Have you any concerns about statistical analyses in this paper?

No

Recommendation?

Accept with minor revision (please list in comments)

Comments to the Author(s)

I would like to thank the authors for addressing my previous comments. I have a few minor new comments listed below that I think should be addressed before publication.

L. 115-117: how did you select these colonies? I'm assuming that it was at random, but you should specify it for clarity.

L. 280-285: the distributions of the R parameter are different for each species but the explanation for this difference is given much later in the result section. Consider adding here a reference pointing to the relevant result section for clarity.

L. 312-314: This is interesting. The absence of an effect of colony size suggests that the STACs may not be driven by social interactions after all. Indeed, higher densities of individuals should lead to higher contact rates and, almost inevitably, to higher activation frequencies and shorter dominant periods (in the limit of the refractory period). I can see two possible reasons for this lack of effect under the hypothesis that the STACs are actually driven by social interaction. First, ants may maintain a roughly constant local density and, therefore, contact rate. I believe that this can be easily tested with the data from this study (e.g., by looking at the distribution of nearest-neighbor distances as a function of colony size, or the average density of the convex hull of the colony). Or, there may not be enough variance in the colony sizes to detect an effect. Note that the methods section only reports the minimum and maximum sizes which may give a false impression that there is actually a large variance in the sample. In addition, an analysis of the effect of colony size on STACS and multi-rhythmicity could also easily be done with the model to demonstrate that this lack of effect is to be expected (or not) under the hypotheses of the model. I think this would strengthen the study but I will defer to the editor the decision to request that additional analysis.

Decision letter (RSOS-211908.R0)

Dear Dr Doering

On behalf of the Editors, we are pleased to inform you that your Manuscript RSOS-211908 "Noise resistant synchronization and collective rhythm switching in a model of animal group locomotion" has been accepted for publication in Royal Society Open Science subject to minor revision in accordance with the referees' reports. Please find the referees' comments along with any feedback from the Editors below my signature.

Please submit your revised manuscript and required files (see below) no later than 7 days from today's (ie 02-Feb-2022) date. Note: the ScholarOne system will 'lock' if submission of the revision is attempted 7 or more days after the deadline. If you do not think you will be able to meet this deadline please contact the editorial office immediately.

on behalf of Professor Roland Bouffanais (Associate Editor) and Kevin Padian (Subject Editor)
openscience@royalsociety.org

Editor comments:

Thanks for your attention to the comments of the reviewers. As you see, one is quite happy, another has a few suggestions that you should be able to address easily, and the third is not happy with some aspects of it but I think you can take those comments into consideration constructively. Please make final modifications and return it to us. Best wishes.

Reviewer comments to Author:

Reviewer: 2

Comments to the Author(s)

The authors have revised the article. However, I do not think that it is yet ready for publication.

Firstly, I have a still concern about gaps between experimental results and simulation results in respect with active cycle periods. Although the authors have stated that “(L384) However, making the agents choose the direction of their next step completely randomly....”, this is not applied to the case of *L. crassipilis*. Furthermore, it is not likely that ants choose the direction of their nest step randomly.

Secondly, the parameter R following an exponential distribution seems to fail to produce multirhythmicity, which is conflict with the experimental fact. Maybe, the mechanism of multirhythmicity observed in *L. crassipilis* is different from the model. The authors have stated that “(L493) ...our model’s primary insights still stand: mobile excitable systems can synchronize when agents have noisy refractory states, and birhythmicity can be diminished through the addition of refractory noise.”. I agree with that. At the same time however, I get impression that the authors try replicating experimental phenomena using proposed model. In such circumstances, I cannot ignore these inconsistencies between simulations and experiments.

As it is already a revised version of the article, I am sorry but I cannot recommend acceptance or minor revisions and therefore I rather suggest to reject the article from publication.

Reviewer: 1

Comments to the Author(s)

The authors have done a great job in addressing my concerns in the revision of the manuscript. Of course, there are some limitations of this study. But, it’s been discussed enough.

Reviewer: 3

Comments to the Author(s)

I would like to thank the authors for addressing my previous comments. I have a few minor new comments listed below that I think should be addressed before publication.

L. 115-117: how did you select these colonies? I’m assuming that it was at random, but you should specify it for clarity.

L. 280-285: the distributions of the R parameter are different for each species but the explanation for this difference is given much later in the result section. Consider adding here a reference pointing to the relevant result section for clarity.

L. 312-314: This is interesting. The absence of an effect of colony size suggests that the STACs may not be driven by social interactions after all. Indeed, higher densities of individuals should lead to higher contact rates and, almost inevitably, to higher activation frequencies and shorter dominant periods (in the limit of the refractory period). I can see two possible reasons for this lack of effect under the hypothesis that the STACs are actually driven by social interaction. First, ants may maintain a roughly constant local density and, therefore, contact rate. I believe that this can be easily tested with the data from this study (e.g., by looking at the distribution of nearest-neighbor distances as a function of colony size, or the average density of the convex hull of the colony). Or, there may not be enough variance in the colony sizes to detect an effect. Note that the methods section only reports the minimum and maximum sizes which may give a false impression that there is actually a large variance in the sample. In addition, an analysis of the effect of colony size on STACS and multi-rhythmicity could also easily be done with the model to demonstrate that this lack of effect is to be expected (or not) under the hypotheses of the model. I think this would strengthen the study but I will defer to the editor the decision to request that additional analysis.

===PREPARING YOUR MANUSCRIPT===

one version should clearly identify all the changes that have been made (for instance, in coloured highlight, in bold text, or tracked changes);

===PREPARING YOUR REVISION IN SCHOLARONE===

-- If you are requesting an article processing charge waiver, you must select the relevant waiver option (if requesting a discretionary waiver, the form should have been uploaded, see 'File upload' above).

-- If you have uploaded any electronic supplementary (ESM) files, please ensure you follow the guidance at <https://royalsociety.org/journals/authors/author-guidelines/#supplementary-material> to include a suitable title and informative caption. An example of appropriate titling and captioning may be found at https://figshare.com/articles/Table_S2_from_Is_there_a_trade-off_between_peak_performance_and_performance_breadth_across_temperatures_for_aerobic_scope_in_teleost_fishes_/3843624.

Author's Response to Decision Letter for (RSOS-211908.R0)

See Appendix B.

Decision letter (RSOS-211908.R1)

Dear Dr Doering,

I am pleased to inform you that your manuscript entitled "Noise resistant synchronization and collective rhythm switching in a model of animal group locomotion" is now accepted for publication in Royal Society Open Science.

on behalf of Professor Roland Bouffanais (Associate Editor) and Kevin Padian (Subject Editor)
openscience@royalsociety.org

Appendix A

Reviewer comments to Author:

Reviewer: 1

Comments to the Author(s)

The authors studied short-term activity cycles in ant colonies and revealed the effects of intrinsic noise and interspecific differences in individual-level behavior on the cycles in a whole colony. This research topic is important for not only collective behavior in animal groups but also complex systems such as neural networks and swarm robotics. The manuscript is written well. However, I have some concerns about the simulation and the result.

*****Response:** We are pleased that the reviewer has a positive opinion of our work. We have made a number of improvements to the manuscript based on the comments we received, and we are hopeful that our updated paper resolves any lingering concerns that the reviewer may have.

Major

- It seems interesting that the simulation model has a bifurcation from one cycle to multirhythmicity based on parameters R and Ω . However, since it is only a simulation with narrow ranges of parameters (e.g., fixed or uniform distributed R , $\Omega = 4050$, $A = 216$), we do not really know the mechanism of these characteristics. Of course, an analytical model for this simulation would be outside of the scope of the current study. However, the authors should explore the model behavior in the simulations by changing the parameters and the distributions. For example, how does the bifurcation diagram shown in Fig. 4A, B in the case of R value following an exponential distribution like the case of *L. crassipilis*? If it cannot produce multirhythmicity, the mechanisms of multirhythmicity observed in *L. crassipilis* may differ from the hypothesis.

*****Response:** We appreciate the reviewer's interest in an additional exploration of our model's parameter space. We have now run more simulations where the refractive parameter R is sampled from an exponential distribution [Lines 400-402; Figure S4]. The results from this extra investigation are now presented as a figure in the supplementary material. When the parameter R follows an exponential distribution, we found that the transition to birhythmicity does not occur. Because of this, we now more explicitly draw attention to the fact that the cause of any potential multirhythmicity or multiple co-occurring collective periodicities in actual colonies is not certain [Lines 458-464]. However, this uncertainty surrounding the collective behavior of real colonies does not diminish the primary findings of our model, which are 1) that synchrony and multirhythmicity are possible in an excitable system that shares many characteristics with real *Leptothorax* colonies including behavioral noise, and 2) that adding uncorrelated noise can reduce the occurrence of multirhythmicity. However, the results of the additional simulations do mean that the nature of the different oscillation frequencies seen within and between colonies remains enigmatic. In line with this, we also reiterate the novel findings of our model in the discussion [Lines 423-427; 492-495].

- Mobility of ants is explicitly implemented in the simulation model. However, it would not play an essential role in the colony rhythms because it is just a random walk. For example, I guess that the effect of the mobility would be the same as assuming that the probability that inactive becomes active via interactions is proportional to the number of currently active individuals without considering space. Please discuss the difference between the actual and simulated movements.

*****Response:** The exact nature of the movement paths of agents in the simulation is indeed not an essential part of our model. To accurately model ant movement paths, we would have needed to take into account differences between individual workers. The movement paths and nestmate interactions of individual *Leptothorax* ants (and their close relatives *Temnothorax*) are partly determined by a number of forces including dominance hierarchies (subordinate ants will avoid moving close to dominant ants) and spatial fidelity zones (individuals ants will spend more time walking in some parts of the nest than others). Both of these factors make interactions between some individuals more likely than would be the case if interactions were completely random. The current model ignores these details, and a nonspatial model where pairs of randomly chosen ants interact with one another would likewise forgo such details. Modeling these factors would add unnecessary complexity to the model and is beyond the scope of the study. Fortunately, the research goals of our agent-based model are not contingent on the specifics of how real ants walk inside their nests because the primary purpose of our model was to investigate how noise affects synchronization and multirhythmicity in excitable systems that are similar to (but not identical with) *Leptothorax* colonies. Although it is not essential to our empirical or computational work, we have now run a new analysis to evaluate the effects that different movement patterns have on the dominant collective period of simulated colonies [Lines 249-257; 381-386]. We now include this analysis in the supplementary material and mention in the main text how this simplification is a limitation of the present work that could be addressed in future research [Lines 478-479; 481-488].

- line 109: When did you start and finish filming? I am concerned about an effect of circadian rhythms. There is a possibility that the multirhythmicity comes from the difference of activity in morning, afternoon, and night. There are no circadian rhythms in these species? Did you confirm that by 35 hours observation?

*****Response:** The 35-hour colony recordings were started in the morning between about 10:30-11:30AM and filming of the 9-hour colony recordings were started in the afternoon between about 12-4pm. Therefore, time of day/circadian rhythms cannot account for the greater variability in the dominant short-term activity cycle oscillation periods in *L. crassipilis* compared to *L. sp W*. However, our new quantitative analysis of multiple rhythms in the 35-hour time series shows that there are indeed multiple co-occurring activity rhythms in both *L. sp W* and *L. crassipilis* [Lines 177-184; 315-333]. As we mention in our responses to other comments in this document, we now point out that whatever is causing the multiple rhythms in real colonies is unknown [Lines 458-464]. There are a number of hypotheses that we could consider, including the possibility that the period might be linked to some extent to circadian rhythms. However, in a currently unpublished manuscript that we are preparing for publication, we did not find any link between activity cycle period and time of day in colonies that were kept in constant lighting and temperature conditions (i.e., the same setup used in this experiment), so we don't think that explanation is likely. Furthermore, not all the colonies that we have 35-hour time series for exhibited multiple periodicities. For example, in the revised manuscript we now show that the *L. crassipilis* colony L4 maintained a consistent 2.6-hour period over 35 hours [Lines 327-330]. In any case, we don't really know the mechanisms causing real colonies to have these kinds of complex rhythms, but the observation is a novel result so we feel that it should be presented in this paper.

Minor

- line 63: though -> through

***Response: We have now fixed this mistake. [Line 65]

- line 236: were -> where

***Response: This has now been fixed. [Line 295]

- lines, 249, 251, 264, 266, 268, 269, 297, 300: Please clarify S.D. or S.E.

***Response: This information was included in the supplementary material, but we have now moved these details so that they instead appear in the methods section of the main text "*All time series summary data are presented as average \pm standard deviation.*" [Line 176]

- line 275: (-> :

***Response: We have now corrected the misplaced brackets. We also replaced the colon separating "L. sp W" and "GLMM" with a dash "-" to help improve the clarity of how we grouped our results. The colon separating "L. crassipilis" and "GLMM" has likewise been replaced with a dash "-". [Line 350-351; 354-355]

- figures: Please unify upper/lower case of alphabets a, b, c,...

***Response: All callouts to specific figure panels now use uppercase letters.

Reviewer: 2

Comments to the Author(s)

The authors examined the activity patterns of ant workers using two different species and found some differences between them, which were the distributions of colony oscillation frequencies, the distributions of worker inactivity durations and so on. The authors also developed a simulation model to explain collective oscillations using the empirical data. This paper is potentially interesting. However, before the paper is considered for publication, there are questions that must be answered.

***Response: We are grateful for the reviewer's helpful comments and suggestions. We are also pleased that the reviewer regards our work on ant activity cycles as interesting, and we have worked to revise our manuscript to resolve the points raised in their review.

Major concerns:

I have a concern about gaps between experimental results and simulation results in respect with active cycle periods. Please see my comments below. Some of them may be related to this issue.

I am not sure whether density of artificial ants in the simulation field is consistent with that in ant experiments. Can you clarify it for me? Linked to this, is there any possibility that durations of activity cycles depend on ant densities?

*****Response:** We now explain that the density of agents used in the simulations is biologically reasonable [Lines 265-274], and we clarify that investigating the effect of agent density on collective activity cycles was not a goal of this study as we were primarily interested in exploring the occurrence of noise, refractory states, and multiple rhythms, both in real colonies and in an idealized model of the ants' activity cycles [Lines 88-90; 265-267]. We also mention in the current manuscript that colony size was not correlated with cycle period in either *L. sp W* or *L. crassipilis*. [Lines 312-314]

You have stated that "A stimulation event was defined as the moment an active and inactive agent became 1 patch apart." What happens if an active agent and an inactive agent occupy the same patch with each other? Does a stimulation event occur at that time?

*****Response:** We have clarified that a patch in the simulation is a square that is 1 by 1 length units long [Lines 240-241]. The units of the simulation are arbitrary length units, but as we now explain in the methods, 1 length unit in the simulation corresponds to approximately 3mm in real life because of how far the agents move each time step and because of the distances that interactions occur at within the simulation [Lines 267-270]. We also now explain that a stimulation event occurs if the distance between the coordinates of a pair of agents is less than 1 unit (i.e., they are at least 1 "patch" apart) [Lines 242-243]. Active agents move 1 unit every time step, and because a patch is a 1x1 unit square, they thus move between adjacent patches every time step that they are walking. We also added a sentence that further explains the situation of overlapping agents "*If two inactive agents occupy the same patch and one of them becomes active, this would therefore also qualify as a stimulation event if the ants were within 1 length unit of each other.*" [Lines 244-246].

In a supplementary file, you have stated that "The mean for parameter S was determined for both species by taking the average value of isolated individuals average IBI values". Also, according to a supplementary file, the parameter A is set as a fixed value in both species. However, time series analysis reveals that activity rhythms of isolated individuals are little bit complex and appear to depend on individuals. Moreover, CV is quite different between two species. Don't you simplify these parameter settings? Is it correct that simulation results adequately reproduce results found in figure 2a, 2b, 2c and 2d? Is there any possibility that parameter settings may be related to mustaches regarding active cycle periods between experiments and simulations?

*****Response:** We agree that the current model involves many simplifications, but the present model is only meant to be an abstraction. As such, the model incorporates several important features of short-term activity cycles in ants, and we are aware that it neglects other potentially relevant features. The present model does however replicate the emergent synchronization of individuals and shows that uncorrelated noise can have a beneficial effect on multirhythmicity in this kind of excitable system. We discuss the possibility that including heterogeneity of individual behavior in a model could at least partially account for the discrepancies between our model and the empirical data from colonies [Lines 478-479]. In our revised manuscript, we have also expanded the portion of the discussion that mentions the limitations of our work [Lines 481-488].

Minor comments:

-Do single patches approximate to the body size of single ants?

*****Response:** Indeed. Because simulated ants interact with other ants based on if they are less than one unit/patch apart, and because simulated ants occupy a single patch, agents can be thought of as being functionally the same size as patches [Lines 267-270].

-What is a definition of multirhythmic behaviour?

*****Response:** We described multirhythmicity in the introduction of the paper, but we have now rephased our sentence to improve the clarity of our definition. [Lines 87-88]

-Regarding figure 1D and 1E, I am not sure which figure is L sp. W's. Can you fix that?

*****Response:** We have now added arrows that point from the pictures of each species to the corresponding time series columns. [Figure 1]

-You have stated that "The aggregate data from L. sp. W is right skewed and unimodal, but the distribution..." Can you plot a best fit unimodal or exponential law for individual species in figure 2F?

*****Response:** We have now plotted best fit log-logistic and exponential fits for the L. sp W and L. *crassipilis* data, respectively [Figure 4F]. Note that because of the addition of new figures in our revised manuscript, figure 2 from the previous manuscript is now figure 4 in the revised manuscript. We have also updated the figure caption to reflect this addition [Lines 715-716].

Reviewer: 3

Comments to the Author(s)

General comments

This manuscript presents a study of the individual and collective activity patterns of two species of *Leptothorax* ants. The stated objectives of the study are to (1) add to the corpus of already existing data on the topic to better understand interspecific differences in individual and collective activity patterns, (2) investigate the existence of multirhythmicity in the activity pattern of ant colonies, and (3) develop a data-driven model to explore the role of individual-level "noise" on the collective activity output of an ant colony. The premises of this study are extremely interesting; indeed, the question of the origin of activity patterns in social groups is of great interest to any field interested in the emergent properties of collective systems. However, I think that this particular study falls short on each of its stated objectives, and while it certainly shows good potential, it also requires additional work to be complete.

*****Response:** We are happy to hear that the reviewer finds the subject of our work interesting, and we appreciate the comprehensive assessment that they have provided. As the reviewer nicely summarizes,

our study had three objectives. Although we definitely agree that much more interesting results can be obtained on the subject of short-term activity cycles, we feel that such findings will need to be left for subsequent studies that use more sophisticated technologies to track the individual-level behavior of all ants in a colony simultaneously. As we explain in our other responses below, the data from the current study is insufficient to fully investigate all of our desired objectives to their fullest potential. However, in this revision, we have attempted to 1) resolve many of the concerns raised by the reviewer by improving the analysis of our data, and 2) more clearly state both the novelty and limitations of our current work in the introduction and discussion sections. We view the principal novel contributions of our study's empirical data as being the following: 1) the observation of long (> 50-min long) oscillation periods in *Leptothorax* (previously unknown from this genus) and a demonstration that typical dominant collective oscillation frequencies can differ between *Leptothorax* species, 2) the observation that multiple periodicities can be present in a single colony's time series record (a finding that is bolstered by a new quantitative analysis in our revision), and 3) providing evidence that ants have refractory states (our new analysis now controls for the possibility that a threshold number of interactions causes ants to become active) and that the two species studied both achieve synchrony despite differences in the refractory/individual-level behavior of ants. We view the principal novel contributions of our study's modelling work as being: 1) simulated ants can synchronize to a common rhythm despite probabilistic refractory state durations, and 2) that uncorrelated noise can reduce multirhythmicity in excitable systems. We believe that the empirical results offer interesting new observations on the natural history of short-term activity cycles in *Leptothorax* and that the model's results illustrate an intriguing effect of noise in a system that shares important similarities with real *Leptothorax* colonies. It is our opinion that these results, though they may not be top-tier research breakthroughs, are indeed novel and do merit publication.

For (1), the authors state that "empirical data are scarce and limited to just a handful of species", mainly ants of the genera *Temnothorax* and *Leptothorax*. They then proceed to present a study on two more species of... *Leptothorax* ants, with similar colony sizes and organization as species in previous studies. To be clear, I am not suggesting that the data collected as part of this study is useless (they are not); but that if one of the objectives of the study was to, indeed, better understand interspecific differences, then the authors should have collected data on a wider array of species, ideally with different colony organization and across orders of magnitude in colony sizes. Simply comparing two close species does not allow for identifying much of the behavioral and ecological determinants of collective activity patterns. I suggest that the authors simply drop that stated objective from their introduction since they are not really pursuing it and the other objectives are interesting enough on their own.

*****Response:** We actually intended a much narrower conception of interspecific variation here. Specifically, we were referring to the variation seen between species in the typical *length* of collective activity cycles. The observations reported in this paper from *Leptothorax crassipilis* show that it is the first member of its genus known to have long (i.e., greater than 50-min) collective activity cycles. A collective-level intraspecific difference thus exists between *Leptothorax* ants in typical cycle lengths. We wanted to see if our version of a model of short-term activity cycles could replicate the observed collective difference when individual-level species differences were incorporated. Ultimately, our model simulations were not able to produce these very long cycles, but we mention in the discussion how this is a limitation of the modeling paradigm we used, and that future work is needed [Lines 464-476; 490-492]. However, we feel that it is worthwhile to mention that we collected data on individual-level

behavior in these two particular species precisely because they exhibit a difference in the dominant collective frequencies at which they tend to oscillate. The choice to utilize two species that are closely related and who possess similar social organization and colony sizes was deliberate as it facilitated using the same methods on both species. To clarify this in the paper, we have therefore modified the first mention of “interspecific differences” in the introduction to explicitly refer to “interspecific differences in cycle frequency” [Line 89]. We have also deleted the phrase “and limited to just a handful of species” [Line 72] to avoid potentially misleading readers that the study is a broad survey of newly observed activity cycles from many diverse species.

For (2), the authors use the fact that there is more variability in the distribution of activity peaks in one of the studies species to suggest that there is "a potential for multirhythmic collective cycles". This is at best a far-fetched conclusion from the data. The authors did not even take the time to first check whether this increased variability was the product of a non-random process or not (for instance, using a modified version of the approach they used to do exactly that for individual workers; see detailed comment below). By the look of the period distribution in figure 1B, I would not be surprised to find that it is just the result of a random Poisson process with a minimum delay. If that was the case, then the authors could conclude that there certainly is NO multirhythmicity in their data instead of allowing doubt to linger on their conclusions. And if it is not random, then the authors could probably keep speculating about the presence of multirhythmicity in their data but should be very clear that it is nearly impossible to formally demonstrate it given their data.

*****Response:** The sentence where we say "a potential for multirhythmic collective cycles" was unfortunately ambiguous, and we have now removed it [Line 312]. We did not intend to portray the greater variability in the dominant periods of *L. crassipilis* colonies (when compared to *L. sp W*) from the 9-hour recordings as evidence for multirhythmicity. Instead, our evidence for multirhythmicity in the manuscript came from our reference to figure 1d (a 35-hour time series), but our point was communicated poorly. In any case, as the reviewer mentions in this comment and elsewhere, our claim of multirhythmicity was not based on a quantitative, objective analysis. To rectify this, we have now included a new analysis of the 35-hour colony recordings using Lomb-Scargle periodograms just as we do for the simulated time series from our model (after taking care to detrend the time series of any long-term trends). This allows us to examine the empirical time series more rigorously for the presence of multiple periodicities [Lines 177-184; 315-333]. We now report that multiple distinct oscillation frequencies are clearly present in the power spectra of individual time series. For example, colonies L11 and L16 both oscillate with a period of around 1-1.5 hours, but the fluctuations in their activity time series also follow longer cycles of around 4 hours [Figure 2]. We also now show that these longer rhythms can also occur in both *L. sp W* and in *L. crassipilis* as well. In our data, the dominant oscillation period of *L. sp W* colonies was always nearly 20 min, and the dominant oscillation period of *L. crassipilis* colonies was quite variable. However, this new analysis reveals that long and short-term rhythms can also co-occur in the individual time series of both species even though the signal of longer, >50-min rhythms are less detectable in *L. sp W*. Since the length of all the longer time series is 35 hours, it is possible for the presence of frequencies in this range (circa. 20 min to 3.5 hours) to be accurately detected.

We now emphasize in the discussion that the cause of the multiple periodicities in real colonies is unknown, and it is also unknown if the behavior seen in the long recordings qualifies as genuine

multirhythmicity where colonies repeatedly switch between distinct, stable frequencies [Lines 458-464]. The spectral analysis however does allow us to conclude that multiple rhythms can at least occur concurrently in real *Leptothorax* colonies (in both species). The time series plots in the new figure 2 also give the impression that the strengths of the various rhythms appear to fluctuate over time [Lines 324-327]. This second finding is admittedly far from definitive, and we point out this uncertainty in the discussion [Lines 458-464]. Because our new analysis uncovered evidence that both species had multiple co-occurring rhythms in the 35-hour recordings, we have also revised parts of the abstract [Lines 32-34], the results [Lines 316-318], and the discussion [Lines 406-408; 436], to state that the *dominant* collective oscillation frequency is more variable in *Leptothorax crassipilis* colonies, but that longer rhythms can occur alongside faster rhythms in both species.

Finally, for (3), the proposed modeling study is very incomplete and does not really provide any new reliable insight into the observed activity patterns. First, the authors do not consider in their model and data analysis at least one alternative mechanism (stimulation threshold) that could explain the shape of their data. Modeling is not about finding a mechanism that matches the data reasonably well but about finding the most likely mechanism given the data and existing knowledge. Then, the authors, noticing that their model does not match their data, do not attempt to identify the possible cause for that large mismatch. An obvious first suspect (besides the alternative mechanism discussed before) is that, while the authors were careful to measure in their data several of the model's parameters, they completely ignored the interaction pattern between the agents and set the associated parameters to generic values without regards for the actual movement behavior of their ants. Yet, it is well-established that the interaction pattern between agents is critical to understanding the output of a collective dynamical system and it should have been investigated more carefully here. Finally, while the entire discussion about the role of "noise" on the collective output of the system is interesting in itself, it is not exactly new in the fields of collective behavior and complex systems (see detailed comment). It may be more novel in the specific context of understanding activity patterns in ant colonies but, because the model fails to convince that it is a good approximation of the observed data, the interest of the entire discussion is greatly reduced. What do we really learn from that model about the emergence of collective activity patterns in ants? Not much because we cannot really trust it is a good representation of the system in the first place.

In conclusion, I cannot recommend the publication of this manuscript at this stage. The authors have done a great deal of work collecting valuable and interesting data but there remains a significant amount of work to finish its analysis and turn it into a reliable model that can generate new insights into the hows and whys of activity patterns in social systems. The collected data have a lot of potential for interesting findings, and I hope the authors will take the time to exploit it fully.

*****Response:** We are pleased that the reviewer finds our work to have value and potential, but we feel that a few of their recommended analyses fall outside the scope for the current study and cannot be effectively performed using the data that we presently have. Regarding the walking paths and interaction patterns of agents in our model in particular, we believe that accurately modelling the interaction patterns of *Leptothorax* would be a vast and complex endeavor and that it would be insufficient for us to simply track the typical paths of ants walking inside the nest. An accurate model of how *Leptothorax* and *Temnothorax* walk inside their nests would need to include the fact that some ants

prefer to spend more time in some portions of the nest than others (i.e., Spatial fidelity zones) (Sendova-franks and Franks 1995), and the fact that dominance hierarchies can cause subordinate ants to avoid going near dominant gynes and workers (Cole 1981). All of these elements could be important to building a realistic model of how *Leptothorax* ants walk and interact inside their nests, yet measuring such elements with the data we currently have and incorporating them into the model would not be practical nor feasible because we would need detailed tracking data from multiple individuals living inside their colonies over long periods of time. We now explain this limitation more effectively in the discussion [Lines 481-488].

However, in order to shed some light on how different types of walking paths could influence our model's behavior, we have now run some additional simulations where the amount of stochasticity in the random walk of agents was set to different levels [Lines 249-257]. We did this by running simulations where agents adjusted their headings by either ± 5 degrees of their current heading each time step (resulting in straighter paths than the ± 45 degrees used in the main text) or ± 360 of their current heading (a fully random walk). We found that the amount of randomness in the walk of agents did not have much of an effect on the dominant oscillation periods of the resulting time series simulations, except in the case where the model's other parameters were based on the data from *L. sp. W.* In this case, making agents have fully random walks (next heading = old heading ± 360 degrees) resulted in collective oscillations whose dominant period more closely matched those seen in the empirical data [Lines 381-386; Figure S3]. The fact that our new sensitivity analysis did not uncover major effects of increasing randomness in the walking paths of agents on the collective period implies that the discrepancies between the model and the empirical data are more substantial than can be addressed by fine-tuning the diffusion in the random walk of agents. Earlier models of ant Short-term activity cycles also ignore the specifics of the interaction patterns. For example, the model created by Goss and Deneubourg has agents interact by randomly pairing agents each time step (Goss and Deneubourg 1988), the model developed by Solé et al. uses a fully random walk (Solé et al. 1993), and the epidemic-inspired model of Cole uses a compartmental system of ODEs that does not simulate individual-level ant behavior (Cole 1992). The lack of attention to this detail did not detract from these models because their conclusions did not depend on a faithful simulation of how ants walk. Our model similarly ignores the nuances of how *Leptothorax* actually walk. Although this limits what the model can tell us about short-term activity cycles themselves, our model still provides us with the new insights that rhythm synchronization can be achieved among agents that have stochastic refractory states and that adding uncorrelated noise can dampen multirhythmicity. We now explain in the discussion that neither of the two main findings of our model are contingent on the model being a good approximation of how *Leptothorax* actually move inside their nests; they are general conclusions about the behavior of an excitable system that shares *some* relevant features with *Leptothorax* [Lines 492-495].

Detailed comments

l. 60: "there is no external signal that synchronizes colonies" → "there is no evidence of external signal that synchronizes colonies"

***Response: We have now modified the sentence accordingly. [Line 62]

l. 72: I would replace "noise" by "(behavioral) variability" throughout the text. "Noise" is the result of an

observer's imperfect ability to measure a signal in the data and it is not a characteristic of a system (it's a characteristic of the relationship between the system and the observer). "Variability" (i.e., the fact that all parts of a system don't all behave similarly to each other or to themselves over time), on the other hand, is a characteristic of a system and can affect its global outcome ("noise" only affects the observer's perception of the outcome). These concepts are often confused because statistical variance is a combination of the variability of the system (what the observer truly wants to know) and of measurement errors.

*****Response:** We now recognize the ambiguity that our current use of the term "noise" could cause. To avoid confusion with how the term "noise" is used in some fields, we have now more explicitly defined our use of "noise" in the introduction as being "the amount of inherent randomness or unpredictability in the behaviour of individuals" [Lines 74-76]. This definition of noise being intrinsic to the system being observed (and not merely the result of limitations in an observer's ability to perceive a signal) appears to be the standard definition in at least some fields. For example, the concept of *neuronal noise* is explicitly defined as the inherent randomness in the behavior of neurons (both in terms of variability of firing rate and in response to stimulation), and it is not due to measurement error (Destexhe and Rudolph-Lilith 2012; Longtin 2013). Such a conception of noise is also found in theoretical models of both coupled oscillators and neurons; noise is added directly to these systems by modifying the underlying equations to include a random variable or stochastic forcing function. Because neuronal systems share several similarities with the activity of *Leptothorax* ants (spontaneous activity, excitability, etc.) we believe that defining noise in this way is appropriate, and we hope that the new explanation that we have added to the introduction will clarify our use of the term for readers. Another consideration is that the term "behavioral variability" on its own could also be misleading in this context because it is sometimes used to mean consistent inter-individual variation between worker ants (e.g., behavioral castes associated with specific tasks in a colony's division of labor)

I. 145: What is the rationale for choosing this moving average window size? Have you performed a sensitivity analysis on the window size to see how its choice affects the final results? If not, then it should be done.

*****Response:** We now indicate that this window size was chosen to avoid the spurious detection of small, fast oscillations (periods of approx. 1-3 min) in the time series that would be artifacts of the tracking algorithm [Lines 151-153]. We also now include a brief analysis showing how different smoothing window sizes do not greatly affect the determination of dominant periods in our empirical time series using wavelet analysis (apart from filtering artifactual high-frequency noise), and the details are presented in the supplementary materials.

I. 156: was the dominant oscillation period measured using the raw or the filtered data?

*****Response:** We have now clarified that the dominant period of oscillation was determined using the filtered time series data [Line 164].

I. 178: Replace "haphazardly" with "randomly" throughout the text. While the meaning is clear, "haphazardly" also connotes a lack of care or method.

*****Response:** Although we appreciate this suggestion, we would prefer to avoid solely using “randomly” in this context if possible because we do not want to have readers potentially misunderstand our selection here as involving genuine randomness or pseudorandomness (e.g., choosing ants though the use of a pseudorandom number generator). We have therefore exchanged “haphazardly” with “randomly” but specify that this randomness was technically “haphazard” in parenthetical remarks [Lines 194-195; 217-218]. A redundant instance of “haphazard” at [Line 214] has also now been deleted.

I. 182 and onward: This analysis assumes the existence of a refractory period. An alternative model would be that the ants “accumulate” stimulations until it reaches a threshold that triggers activation. Such a threshold model would also give the impression of a refractory period. You can distinguish between both models by looking at (1) whether the probability of activation increases with time since inactivation after controlling for the quantity of social stimulation, and (2) vice-versa, whether the probability of activation increases with the quantity of social stimulation after controlling for the time since inactivation. Both models can in principle explain your data, yet they assume very different underlying physiological mechanisms. Note also that threshold models are commonly used to explain (rightly or wrongly) many aspects of the division of labor in social insects, therefore should be considered in the context of this study.

*****Response:** Thank you for pointing this out. We have now performed an additional analysis of our data (and updated our R code) to see if the likelihood of an ant becoming active after a physical contact can be predicted by the cumulative number of interactions it has received since it became inactive [Lines 201-209]. We show that there is no association [Lines 349-358].

I. 207-208: “When an ant becomes active, it remains so for a fixed duration”. Explain why you chose to do this. It is currently buried in the supplementary material. All modeling decisions should be clearly and completely justified in the main text.

*****Response:** The relevant explanation has now been removed from the supplementary material and now appears in the methods section of the main text. [Lines 285-287] After reading the other recommendation made by the reviewer about our model parametrization section being in the supplementary material, we have also moved this entire section so that it now appears in the main text instead of the supplementary material. [Lines 275-292]

I. 218-220: “The random walk of simulated active ants [...] is similar to other models of random ant movement”. This is a terrible justification that assumes essentially that all ants walk the same way as your ants. Besides, you have all the video data that you need to estimate the actual movement patterns of your ants. Even assuming a simple correlated random walk, you should be able to easily estimate the diffusion coefficient of your ants from tracking (even roughly) a few minutes of your video footage. You can then fit the movement parameters of your model to reproduce that diffusion coefficient. Given that your model relies on correctly reproducing the interaction rate between active and inactive ants, this step is absolutely critical.

*****Response:** As we explain in one of our responses to an earlier general comment above, we do not believe that a precise description of the interaction and walking patterns of individuals are essential to the research questions that we are asking in the present paper. However, to provide some sense of how

different amounts of randomness affects the collective rhythm of simulated colonies, we do now provide an extra analysis using new simulations where the random walk of agents was varied [Lines 249-257; 381-386].

l. 231-232: "Using the empirical data collected from individuals to parametrize our model (supplementary material)". How the parameterization is achieved is a critical part of the modeling process and should not be relegated to the supplementary material.

***Response: The entire section about model parametrization has now been transferred from the supplementary material to the methods section. [Lines 275-292]

l. 254: "indicating a potential for multirhythmic collective cycles". This is a very far-fetched conclusion. Multirhythmicity is characterized by the coexistence of two or more distinct oscillatory states and can be very difficult to formally demonstrate, especially with the small number of observed bursts (relative to the length of each series that is) and the non-stationarity of the time series. You could be simply witnessing irregularly spaced bursts of activity. You can test this formally more easily using the same approach that you used for individual ants (however, you may first want to "normalize" the IBI by subtracting the value of the smallest IBI from all IBIs to account for the "cooling down" time after each peak that constrains the minimum possible IBI that the system can create).

***Response: As we explain in detail in our response to another one of the reviewer's comments, we now include an analysis using Lomb-Scargle periodograms in order to more rigorously investigate the presence of multiple oscillation frequencies in the colony-level time series [Lines 177-184; 315-333]. The new analysis provides evidence that colonies (from both species) can exhibit multiple periodicities in a single time series. We also now make it clear in the discussion that although this demonstrates a previously unknown feature of short-term activity cycle patterns in *Leptothorax*, this is not necessarily multirhythmicity *sensu stricto*. We also stress that the cause of the multiple rhythms in real colonies is not necessarily the same as the cause of the multirhythmicity that we detected in the model simulations [Lines 458-464].

l. 273-onwards: See my comment above. These analyses should be done while controlling for the amount of social stimulation received by the ants to account for an alternative model in which an ant activation depends on the amount of social stimulation it receives and not on the existence of a refractory period (and vice versa).

***Response: These results have now been updated [Lines 349-358] based on our new analysis [Lines 201-209] that controls for the number of interactions received by inactive ants.

l. 320: "patters" → "patterns"

***Response: This has now been fixed. [Line 408]

l. 335-336: "However, to our knowledge, our work is the first spatially explicit formulation of a model that combines agents that are mobile, refractory, and which activate stochastically". I am a bit confused by this statement. There are plenty of examples of models with these properties (e.g., see the firefly model in the Netlogo software that the authors used in this study). There are also several swarm

robotics implementations of such models (again, following the firefly example). Are you saying that this is the first time that this type of model was used in the specific context of ant activity cycles? Even then, the old cellular automata model from Solé in the early 1990s (reference #20 in the manuscript) has all these properties (the stochastic refractory period is implicit in their self-activation function). I would either moderate this claim or make it clearer what is truly new in the proposed model because I do not really see it.

*****Response:** We have now deleted this sentence and several sentences around it since it does not contribute much to the discussion [Line 418]. A key difference between our model and the model of Solé and his colleagues is in the nature of the refractory state. In both models, ants can activate stochastically (e.g., the self-activation of the Solé model). However, in our model the refractory state of agents specifically refers to the times when agents ignore nestmate stimulation. This kind of refractory state is present in the model of Goss and Deneubourg (Goss and Deneubourg 1988), but, unlike our model, agents in that model cannot spontaneously activate during their refractory state, and the duration of the refractory state is not stochastic. As far as we can tell, *inactive* individuals in the Solé et al. model do not have an analogue to our refractory parameter. The only mention of a refractory state in the Solé model is in regards to the fact that agents cannot spontaneously increase their activity level while they are already active (Solé et al. 1993). The research goals of all previous models of short-term activity cycles in ants were also quite different from the goals of this paper. The earlier papers were primarily concerned with establishing the possibility of synchronous and rhythmic collective activity in agents that had either stochastic or chaotic self-activation. Indeed, neither the Solé model nor the earlier Goss and Deneubourg model study the effects that different levels of noise in refractory states can have on synchronization or multirhythmicity. As we now point out in the discussion, the behavior and predictions of the Solé model (in addition to our own model) are incompatible with some of our empirical results [Lines 469-474]. Since our model fares no better in producing very long periods, it is clear more studies are needed to fully understand the short-term activity cycles of *Leptothorax*.

l. 342: "The contribution of noise to the collective behaviour of social insects has received less attention than heterogeneity". Again a very confusing statement. Studying the role of heterogeneity in collective behavior is relatively recent actually (15-20 years old) but studying the role of "noise" in collective behavior has been pretty much constitutive of all modeling work since the early 1980s at least (one of my favorite papers on the topic is Deneubourg J-L, Pasteels JM, Verhaeghe JC. Probabilistic behaviour in ants: A strategy of errors? J Theor Biol. 1983;105: 259–271. doi:10.1016/S0022-5193(83)80007-1). Please reformulate this sentence to make it clear why you think "noise" is understudied in the field of collective behavior.

*****Response:** We have modified this sentence to specify that we are speaking in the context of synchronization [Lines 423-427]. Indeed, the roles of both noise and heterogeneity in social insect collective behavior have been previously studied, but the effects of noise (especially uncorrelated noise) on the social synchronization of activity are not well studied. Our revised paragraph makes this contrast clear.

l. 376: "The simplicity of our model results in some limitations which should form the subject of future work". The main issue in my opinion is that your model does not attempt to reproduce the interaction

pattern between the ants. That alone may be enough to explain the discrepancies between the model and the experiments.

*****Response:** The interaction pattern between ants (along with several other factors) may very well be crucial to fully explaining short-term activity cycles. However, as we mention in some of our previous responses, carefully investigating that question is not the purpose of the present study. The significance of our current model is that it shows that a system that possesses many of the same fundamental characteristics as short-term activity cycles in *Leptothorax* (namely the spontaneous activation of individuals, excitability, mobility of individuals, and the emergent synchronization of individual-level activity patterns) is capable of multirhythmicity and that uncorrelated noise added to the refractory states of individuals can reduce multirhythmicity in the model simulations.

I.394: "The present study does not resolve a central enigma surrounding STACs: why do they exist?" If this is a central enigma, why not testing it with your model? After all, this is what a model is for: testing hypotheses that are difficult (even impossible) to study with traditional scientific approaches. This feels like a missed opportunity to actually do something fundamentally different from previous work on the topic.

*****Response:** The question of potential functional implications of short-term activity cycles in ants is indeed something we are pursuing. However, this question is beyond the scope of the present work as we did not seek to test any hypotheses about either the function or adaptive value of short-term activity cycles. As we explain in some of our other responses in this document, the novelty of our current manuscript lies in the new information that we uncover about the existence of long (>50-min) short-term activity cycle rhythms in *Leptothorax* species, that the individual level behaviors leading to synchronization are different between *Leptothorax* species, and that our model shows that synchronization in mobile excitable media is not inherently diminished by the addition of uncorrelated noise.

Figure 1D: Please, remove the blue (slow) and green (fast) segment indicators. The data does not allow you to conclude that there is multirhythmicity in your data and these indicators are misleading the reader as they are the product of a personal belief and not of an unbiased method.

*****Response:** These subjective brackets and the long time series have now been removed from this figure [**Figure 1**]. The newly added figures 3 & 4 now depict more examples of the long colony recordings and depict a quantitative assessment of multiple co-existing periodicities in the long colony recordings using Lomb-Scargle periodograms [**Figure 3; Figure 4**].

References

Cole BJ (1981) Dominance Hierarchies in *Leptothorax* Ants. *Science* 212:83–84. <https://doi.org/10.1126/science.212.4490.83>

Cole BJ (1992) Short-term activity cycles in ants: age-related changes in tempo and colony synchrony. *Behav Ecol Sociobiol* 31:181–187. <https://doi.org/10.1007/BF00168645>

Destexhe A, Rudolph-Lilith M (2012) Neuronal Noise. Springer US

Goss S, Deneubourg JL (1988) Autocatalysis as a source of synchronised rhythmical activity in social insects. *Ins Soc* 35:310–315. <https://doi.org/10.1007/BF02224063>

Longtin A (2013) Neuronal noise. *Scholarpedia* 8:1618. <https://doi.org/10.4249/scholarpedia.1618>

Sendova-franks AB, Franks NR (1995) Spatial relationships within nests of the ant *Leptothorax unifasciatus* (Latr.) and their implications for the division of labour. *Animal Behaviour* 50:121–136. <https://doi.org/10.1006/anbe.1995.0226>

Solé RV, Miramontes O, Goodwin BC (1993) Oscillations and Chaos in Ant Societies. *Journal of Theoretical Biology* 161:343–357. <https://doi.org/10.1006/jtbi.1993.1060>

Appendix B

Reviewer comments to Author:

Reviewer: 2

Comments to the Author(s)

The authors have revised the article. However, I do not think that it is yet ready for publication.

Firstly, I have a still concern about gaps between experimental results and simulation results in respect with active cycle periods. Although the authors have stated that "(L384) However, making the agents choose the direction of their next step completely randomly....", this is not applied to the case of *L. crassipilis*. Furthermore, it is not likely that ants choose the direction of their next step randomly.

*****Response:** Our sentence about the effect of completely random motion on the model's output was unfortunately ambiguous. We did in fact run simulations where ants chose their next step completely randomly using the parameter sets from both species. This was depicted in figure S3. However, changing the level of randomness in the walk of simulated agents only affected the collective period of simulated *L. sp w* colonies, which is why we mentioned this species in particular. We have revised this sentence so that it now clearly states that the level of randomness in the agents' walks had no effect on the *L. crassipilis* simulations [**Lines 391-395**].

We also certainly agree that ants are unlikely to have a completely random walking pattern. The purpose of including this possibility in our simulations was to evaluate how the walking paths of simulated agents might influence our results, and we did this by looking at the extreme cases where agents would either chose their next step totally randomly or with little change in orientation (± 5 degrees).

While revising this section, we also realized that our notation of saying that agents with a fully random walk adjusted their orientation each time step by "adding ± 360 degrees to their current heading" could be misinterpreted as meaning that agents would always add either 360 degrees or -360 degrees to their heading. Since $360 = -360$ degrees, such a scenario would clearly not result in a random walk because agents would never change directions. Although we already explain earlier in the methods that agents chose a new heading *within* a set range of their old heading [**Lines 238-239**], we have now clarified our notation using brackets that state that in our simulations fully random walkers updated their heading by "adding an integer from the range $\pm [0, 360)$ degrees to their current heading" [**Lines 257-258**]. This updated notation makes it clear that the agents in what we previously referred to as the " ± 360 degree" simulations did in fact have totally random walks. We have thus also updated this bracket notation for how agents updated their headings whenever it occurs in the manuscript and in figure S3 of the supplementary material (e.g., "adding an integer in the range $\pm [0, 5)$ degrees to their current heading") [**Lines 241; 250; 257-260; 389; 392; Figure S3**].

Secondly, the parameter R following an exponential distribution seems to fail to produce multirhythmicity, which is conflict with the experimental fact. Maybe, the mechanism of multirhythmicity observed in *L. crassipilis* is different from the model. The authors have stated that "(L493) ...our model's primary insights still stand: mobile excitable systems can synchronize when agents have noisy refractory states, and birhythmicity can be diminished through the addition of refractory noise.". I agree with that. At the same time however, I get impression that the authors try replicating

experimental phenomena using proposed model. In such circumstances, I cannot ignore these inconsistencies between simulations and experiments.

As it is already a revised version of the article, I am sorry but I cannot recommend acceptance or minor revisions and therefore I rather suggest to reject the article from publication.

*****Response:** As we state in the paper, we were definitely interested in potentially explaining the empirically observed species-level differences with our simple model, but we explicitly discuss the limitations and shortcomings of our model in several places in the discussion. Despite these shortcomings, the model produced a number of interesting results, and several of our empirical observations are also novel. As the reviewer points out, these specific results stand regardless of whether our model reproduces all aspects of the ants' behavior. We believe that our primary results are interesting and should be shared in a publication. Prior models of ant activity cycles also fail to capture some of the empirical observations from our present study, and we view the inconsistencies between the various models and these aspects of our empirical data as directions for further research. We appreciate the reviewer's criticisms here, and we hope that subsequent studies that collect detailed data on the movement patterns and interactions of all individuals in a colony simultaneously will shed more light on the dynamics of synchronization in ant colonies.

Reviewer: 1

Comments to the Author(s)

The authors have done a great job in addressing my concerns in the revision of the manuscript. Of course, there are some limitations of this study. But, it's been discussed enough.

*****Response:** We thank the reviewer for their constructive feedback during the revision process, and we are pleased that they are now satisfied with our manuscript.

Reviewer: 3

Comments to the Author(s)

I would like to thank the authors for addressing my previous comments. I have a few minor new comments listed below that I think should be addressed before publication.

*****Response:** We are pleased that the reviewer's major concerns have been addressed with our revision. We are grateful for the useful suggestions that they provided on the previous version of this manuscript. The updated version of the manuscript now incorporates some minor changes based on the most recent set of comments from the three reviewers.

L. 115-117: how did you select these colonies? I'm assuming that it was at random, but you should specify it for clarity.

*****Response:** Indeed. We now indicate that the *Leptothorax* colonies chosen for the long recordings were done so randomly (i.e., haphazardly) [Lines 116-118]. We also now specify that the colonies that

were filmed for 35 hours were collected in 2018. This was because all of the long recordings were filmed prior to the collection of the rest of the *L. sp w* and *L. crassipilis* colonies in 2019. We now clarify this point [Lines 118-120].

L. 280-285: the distributions of the R parameter are different for each species but the explanation for this difference is given much later in the result section. Consider adding here a reference pointing to the relevant result section for clarity.

*****Response:** We have now added a sentence that states the rationale for why our model used different R parameter distributions depending on the species [Lines 289-291]. As the reviewer suggests, this new sentence also includes a reference to the results section since the reason for the differing distributions in our model becomes clear there. Lastly, we added the word “parameter” to the beginning of the sentence at [Line 293] so that the sentence no longer abruptly begins with the parameter name “A”.

L. 312-314: This is interesting. The absence of an effect of colony size suggests that the STACs may not be driven by social interactions after all. Indeed, higher densities of individuals should lead to higher contact rates and, almost inevitably, to higher activation frequencies and shorter dominant periods (in the limit of the refractory period). I can see two possible reasons for this lack of effect under the hypothesis that the STACs are actually driven by social interaction. First, ants may maintain a roughly constant local density and, therefore, contact rate. I believe that this can be easily tested with the data from this study (e.g., by looking at the distribution of nearest-neighbor distances as a function of colony size, or the average density of the convex hull of the colony). Or, there may not be enough variance in the colony sizes to detect an effect. Note that the methods section only reports the minimum and maximum sizes which may give a false impression that there is actually a large variance in the sample. In addition, an analysis of the effect of colony size on STACS and multi-rhythmicity could also easily be done with the model to demonstrate that this lack of effect is to be expected (or not) under the hypotheses of the model. I think this would strengthen the study but I will defer to the editor the decision to request that additional analysis.

*****Response:** We share the reviewer’s interest in the question of how group size might influence synchronization and activity cycles. The suggested analyses would indeed be useful for investigating this topic, but we feel that they would not exactly be the most appropriate for the current study. Regardless of what we were to find regarding the distribution of nearest-neighbor distances in colonies, it could be possible that ants in larger colonies also alter *how* they interact with nestmates in larger colonies. For example, it may be the case that workers in very large colonies become less responsive to nestmate stimulation if that would reduce overall group coherence. Ultimately, we believe that to adequately research how system size influences synchronization in this system, we would need to conduct additional experiments where we film ant groups of different sizes using individually marked workers. That would allow us to directly answer the question of how much density, social interactions, or other mechanisms might contribute to the coherence of group rhythms. We thus believe that extra analyses on density and group size would be beyond the scope of the present study, and we would prefer not to conduct them here as we do not think that we would be able to reach any well supported conclusions one way or the other based on the nature of our current data

Finally, although the information on colony size is present in the supplementary data available on GitHub, we thank the reviewer for alerting us to the fact that we omitted the information about colony size means in the methods. We have therefore fixed this by now reporting the mean and standard deviation of colony sizes for both species **[Lines 109-110]**.